EMBO
Molecular Medicine

# Increased VEGF-A promotes multiple distinct aging diseases of the eye through shared pathomechanisms

Alexander G Marneros[1,2,*]

## Abstract

While increased VEGF-A has been associated with neovascular age-related macular degeneration (AMD), it is not known whether VEGF-A may also promote other age-related eye diseases. Here, we show that an increase in VEGF-A is sufficient to cause multiple distinct common aging diseases of the eye, including cataracts and both neovascular and non-exudative AMD-like pathologies. In the lens, increased VEGF-A induces age-related opacifications that are associated with ERK hyperactivation, increased oxidative damage, and higher expression of the NLRP3 inflammasome effector cytokine IL-1β. Similarly, increased VEGF-A induces oxidative stress and IL-1β expression also in the retinal pigment epithelium (RPE). Targeting NLRP3 inflammasome components or Il1r1 strongly inhibited not only VEGF-A-induced cataract formation, but also both neovascular and non-exudative AMD-like pathologies. Moreover, increased VEGF-A expression specifically in the RPE was sufficient to cause choroidal neovascularization (CNV) as in neovascular AMD, which could be inhibited by RPE-specific inactivation of Flk1, while Tlr2 inactivation strongly reduced CNV. These findings suggest a shared pathogenic role of VEGF-A-induced and NLRP3 inflammasome-mediated IL-1β activation for multiple distinct ocular aging diseases.

**Keywords** age-related macular degeneration; aging; cataract; NLRP3 inflammasome; VEGF-A
**Subject Categories** Ageing; Neuroscience; Vascular Biology & Angiogenesis

## Introduction

The VEGF-A gene locus has been associated with both neovascular ("wet") and non-exudative ("dry") age-related macular degeneration (AMD) (Yu et al, 2011; Fritsche et al, 2013). VEGF-A is also increased in aqueous humor samples from eyes of patients with neovascular AMD, and a direct pathogenic role of increased VEGF-A is supported by the observed clinical benefit of anti-VEGF-A therapies that can inhibit further progression of neovascular AMD (Funk

et al, 2009; Group et al, 2011). However, chronic use of anti-VEGF-A antibodies can show diminished therapeutic activity with disease progression of neovascular AMD and may induce adverse side effects in the retina, likely by inhibiting neuroprotective functions of VEGF-A or by impairing the choriocapillaris that requires VEGF-A for its maintenance (Marneros et al, 2005; Rofagha et al, 2013). Thus, novel treatment options are needed that target downstream pathways that are activated by VEGF-A and inhibit the pathogenic effects of increased VEGF-A levels [e.g. choroidal neovascularization (CNV)], while maintaining the beneficial effects of VEGF-A (e.g. neuroprotective functions of VEGF-A), and which do not impair tissue perfusion.

Whether increased VEGF-A levels are merely a consequence of aging pathologies or instead a direct pathogenic factor in the progression of aging eye diseases is not known. This distinction has important clinical implications, as defining a pathogenic role of increased VEGF-A signaling in specific aging diseases may suggest novel therapeutic approaches for these diseases that target VEGF-A pathway hyperactivation or downstream cellular pathomechanisms that occur as a consequence of increased VEGF-A levels.

With progressive age, increased hypoxia and oxidative damage occur in many tissues, which are both inducers of VEGF-A expression in the retinal pigment epithelium (RPE) (Klettner & Roider, 2009; Byeon et al, 2010). As VEGF-A itself can induce reactive oxygen species (ROS) in cells (Monaghan-Benson & Burridge, 2009; Marneros, 2013), we hypothesized that increased VEGF-A may further promote oxidative damage in tissues in a progressive age-dependent manner and thereby exacerbate the manifestation of aging pathologies.

For example, VEGF-A induces ROS in endothelial cells in a NADPH oxidase-/rac1-dependent manner that lead to autophosphorylation of the VEGF-A receptor Flk1 and downstream activation of AKT and ERK, resulting in increased endothelial cell migration and proliferation, and antioxidants inhibited the mitogenic effects of VEGF-A in endothelial cells (Abid et al, 2000; Colavitti et al, 2002; Monaghan-Benson & Burridge, 2009). While VEGF-A can increase ROS, it also induces the expression of the antioxidant enzyme SOD2 by a rac1-regulated NADPH oxidase-dependent mechanism, which may serve as a compensatory mechanism to reduce overall oxidative stress in cells (Abid et al, 2001).

Importantly, increased ROS generation has been linked to common age-related eye diseases, including senile cataracts and

1  Cutaneous Biology Research Center, Massachusetts General Hospital, Charlestown, MA, USA
2  Department of Dermatology, Harvard Medical School, Boston, MA, USA
   *Corresponding author. Tel: +1 6176437170; Fax: +1 6177264453; E-mail: amarneros@mgh.harvard.edu

AMD. Environmental risk factors (e.g. cigarette smoking) that contribute to oxidative stress and hypoxia have also been associated with these aging eye diseases (Klein *et al*, 2005).

In the lens, the extent of oxidative stress plays a central role for the development of senile cataracts (Berthoud & Beyer, 2009). For example, reduced activity of the antioxidant enzyme SOD1 has been reported in human senile cataracts (Rajkumar *et al*, 2013). Moreover, mice deficient in Sod1 have increased superoxide radicals in their lenses, decreased levels of the antioxidant glutathione, and develop cataracts earlier and with higher frequency (Olofsson *et al*, 2009, 2012), while overexpression of SOD1 in the lens prevented $H_2O_2$-induced cataracts (Lin *et al*, 2005). Notably, ROS have been reported to further increase VEGF-A expression in lens epithelial cells (Zhang, 2010; Neelam *et al*, 2013), but whether increased VEGF-A promotes cataract formation is unknown.

In the retina, increased oxidative damage has been linked to the progression of AMD and experimental models have recapitulated aspects of AMD as a consequence of oxidation-induced pathomechanisms (Hollyfield *et al*, 2008). Similarly as in lens epithelial cells and other cell types, oxidative stress can directly induce VEGF-A in RPE cells, and we have shown that VEGF-A itself can further promote ROS generation in RPE cells (Klettner & Roider, 2009; Byeon *et al*, 2010; Marneros, 2013). While increased oxidative damage has been associated with aging diseases in the lens and the retina and leads to increased VEGF-A levels, it remains unknown whether increased VEGF-A further promotes age-dependent increases in oxidative damage in these tissues and exacerbates the manifestation of age-related eye diseases.

Here, we hypothesized that age-dependent increases in hypoxia and oxidative damage induce the expression of VEGF-A, which itself further stimulates ROS generation and thereby promotes aging pathologies in the lens and the retina. As oxidative damage can induce NLRP3 inflammasome activation that leads to processing and secretion of the proinflammatory cytokines IL-1β and IL-18, we also hypothesized that VEGF-A-induced ROS promote aging pathologies of the eye at least in part through NLRP3 inflammasome-dependent mechanisms and that targeting NLRP3 inflammasome components could inhibit aging pathologies that are promoted by increased VEGF-A while maintaining the beneficial effects of VEGF-A for the function of the adult microvasculature. Thus, we tested in genetic mouse models whether a moderate increase in VEGF-A expression (~two- to threefold, as it occurs in response to hypoxia or oxidative stress in aged tissues) leads to increased age-related progressive pathologies in the lens and the retina, and whether genetic targeting of NLRP3 inflammasome components or of its regulators can inhibit the manifestation of these common but distinct aging eye pathologies.

## Results

### VEGF-A is expressed in the lens and the RPE throughout life and increased VEGF-A expression correlates with elevation of markers of oxidative stress

Here, we utilized a mouse model that has ~two- to threefold increased VEGF-A levels as a consequence of insertion of a lacZ cassette into the 3′-UTR of the VEGF-A gene that regulates its gene expression (VEGF-A[hyper] mice), reflecting VEGF-A increases as they occur in aging conditions induced by increased hypoxia or oxidative damage (Miquerol *et al*, 1999; Funk *et al*, 2009; Marneros, 2013). For example, VEGF-A levels in aqueous humor samples from eyes of patients with neovascular AMD are ~twofold increased compared to healthy age-matched controls (Funk *et al*, 2009). This moderate increase in VEGF-A in the eye is sufficient to promote AMD, as anti-VEGF-A antibodies inhibit the progression of neovascular AMD in a subset of patients or in experimental laser-induced CNV (Campa *et al*, 2008; Group *et al*, 2011; Martin *et al*, 2011). Thus, VEGF-A[hyper] mice are a clinically representative animal model that allows us to assess the effects of a pathophysiologically relevant increase in VEGF-A for the manifestation of aging eye diseases, such as AMD.

Notably, mice that are hypomorphic for VEGF-A due to the insertion of a lacZ cassette at a different site within the 3′-UTR of the VEGF-A gene (VEGF-A[hypo] mice) did not show the age-dependent pathologies observed in VEGF-A[hyper] mice, revealing that β-galactosidase expression from the VEGF-A locus does not contribute to these aging pathologies but that they are indeed a consequence of increased VEGF-A levels in VEGF-A[hyper] mice.

As VEGF-A[hyper] mice have a NLS-lacZ cassette in the 3′-UTR of the VEGF-A gene, staining for β-galactosidase can be used to detect cellular VEGF-A expression in these mice (β-galactosidase appears as nuclear staining). We found VEGF-A expression in the lens, retina and ciliary body in eyes of young adult VEGF-A[hyper] mice, while no staining for β-galactosidase (and thus VEGF-A) was found in the corneas of these mice (Fig 1A–F).

We found that the RPE is a major cell type in the posterior eye to express VEGF-A in the adult, while expression of VEGF-A was also noticed in cells of the ganglion cell layer and the inner nuclear layer of the retina (Fig 1B). This expression pattern was maintained throughout life and strong VEGF-A expression in the RPE was also found in > 2-year-old mice (Fig 1G). We observed ~twofold elevated protein levels of VEGF-A in the RPE/choroids of VEGF-A[hyper] mice and a lesser increase in VEGF-A in the retinas of these mice (Marneros, 2013). This increase in VEGF-A levels was maintained with age and is similar to the reported ~twofold increase in VEGF-A in aqueous humor samples from eyes of patients with neovascular AMD, making these mice a particularly disease-relevant model to investigate the role of increased VEGF-A for the pathogenesis of neovascular AMD (Funk *et al*, 2009).

In the lens, strong VEGF-A expression in the young adult was found in nucleated lens fibers (Fig 1D). Notably, we found in 24-month-old lenses that this VEGF-A expression in nucleated lens fibers is maintained in the aged lens (Fig 1H). Thus, VEGF-A is expressed throughout life in the lens, albeit the function of VEGF-A for the adult lens is not known. VEGF-A mRNA and protein levels were increased ~two- to threefold in lenses of adult VEGF-A[hyper] mice, and this increase was maintained with progressive age (Fig 2A). Mainly the VEGF-A isoforms VEGF-A[120] and VEGF-A[164] and the VEGF-A receptor Flk1 are expressed in the adult mouse lens, while Flt1 is not expressed in the adult lens (Fig 2B).

As we hypothesized that increased VEGF-A may induce oxidative damage with progressive age, we measured lipid peroxidation byproducts as markers of increased oxidative stress in these mice.

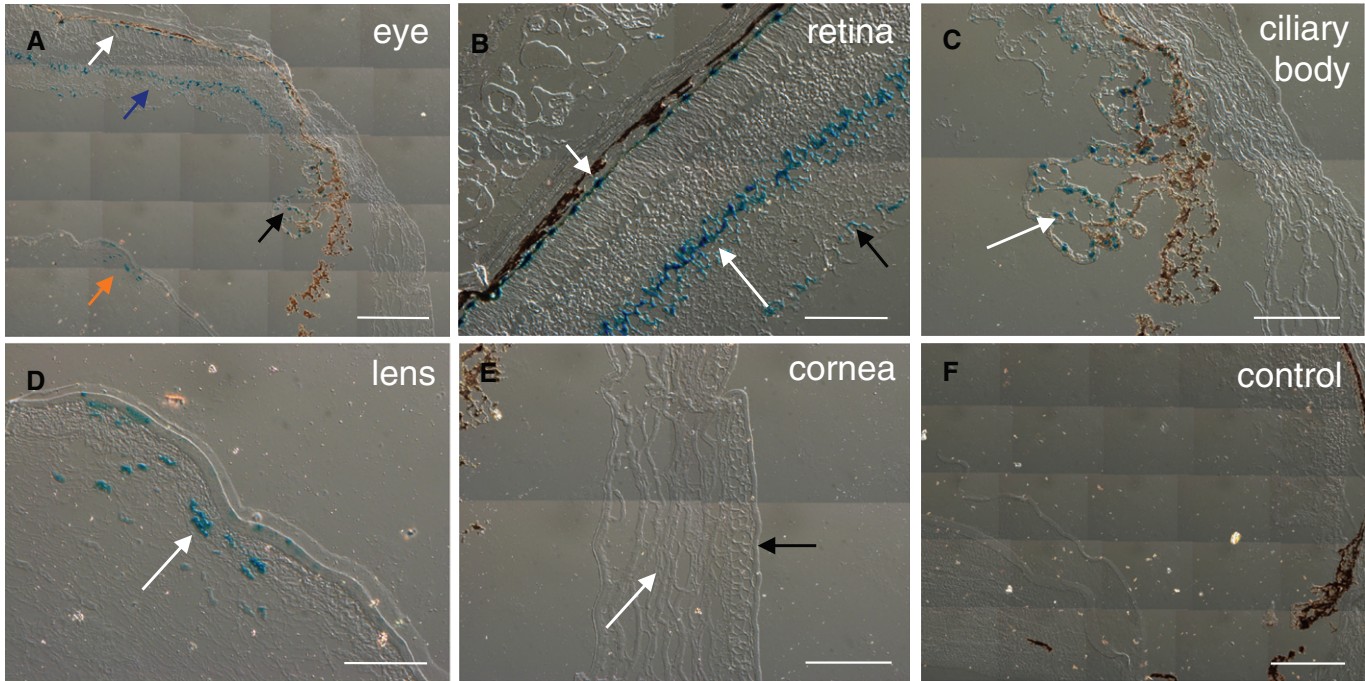

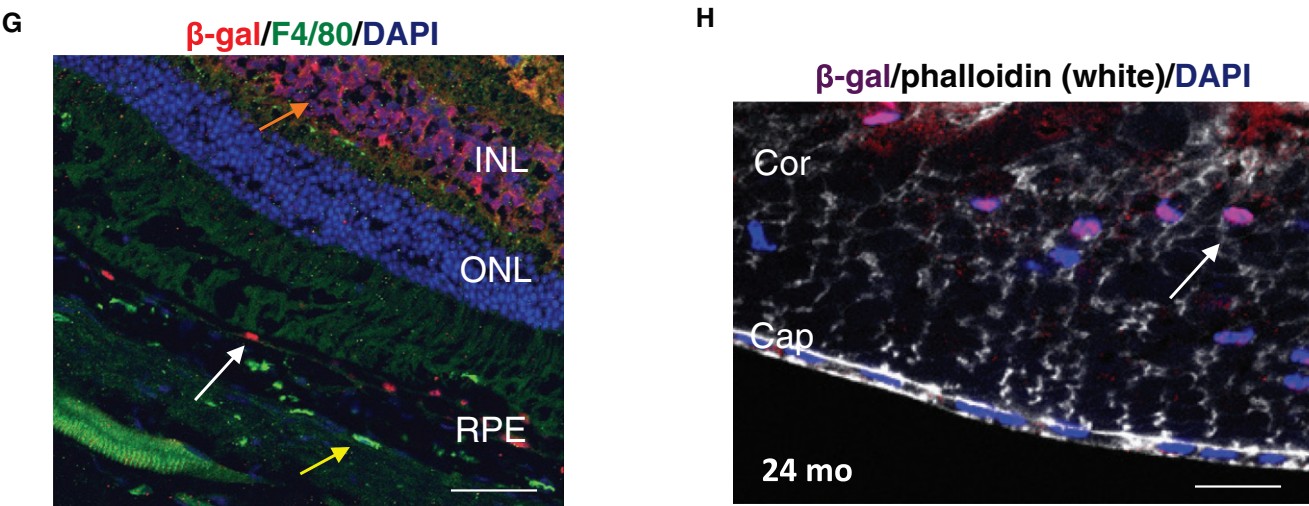

**Figure 1. VEGF-A is strongly expressed in the RPE/retina, the lens, and the ciliary body in the eye, and its expression is maintained with progressive age.**

A–E　Staining for β-galactosidase in 3-month-old VEGF-A^hyper (lacZ/wt) mouse eyes. (A) Low-magnification image shows β-gal[+] cells (in blue) in the RPE (white arrow), retina (blue arrow), ciliary body (black arrow), and lens (orange arrow). Images of (B-E) are higher-magnification images of (A). (B) Strong expression of VEGF-A with β-gal[+] cells is observed in the RPE (short white arrow) and the inner nuclear layer (INL) (long white arrow), while less staining is observed in the ganglion cell layer (black arrow) of the retina. (C) Ciliary body epithelial cells are β-gal[+], similarly as in RPE cells (white arrow). (D) VEGF-A is strongly expressed in nucleated lens fibers in the adult lens (white arrow). (E) No β-gal[+] cells are observed in the corneas of adult VEGF-A^hyper (lacZ/wt) mouse eyes. White arrow shows the stroma of the cornea, and the black arrow shows the corneal epithelium.

F　　No staining for β-galactosidase is observed in 3-month-old control wild-type littermate eyes.

G　　RPE cells maintain strong expression of VEGF-A (β-gal[+] cells, white arrow) with progressive age. Immunofluorescence for β-gal in the retina of a 24-month-old VEGF-A^hyper mouse is shown. Choroidal macrophages (F4/80[+] cells, yellow arrow) show no β-galactosidase expression, while some immunolabeling for β-galactosidase is detected in retinal cells of the inner nuclear layer (orange arrow) of the retina. RPE: retinal pigment epithelium; ONL: outer nuclear layer; INL: inner nuclear layer.

H　　VEGF-A continues to be highly expressed in nucleated lens fibers in the aged lens (shown is a lens of a 24-month-old VEGF-A^hyper mouse; nuclear β-galactosidase staining (arrow) reflects cellular VEGF-A expression). Cor: lens cortex; Cap: lens capsule. DAPI labels nuclei (blue).

Data information: Scale bars, 200 μm (A and F), 100 μm (B, C and E), 50 μm (D and G), 25 μm (H).

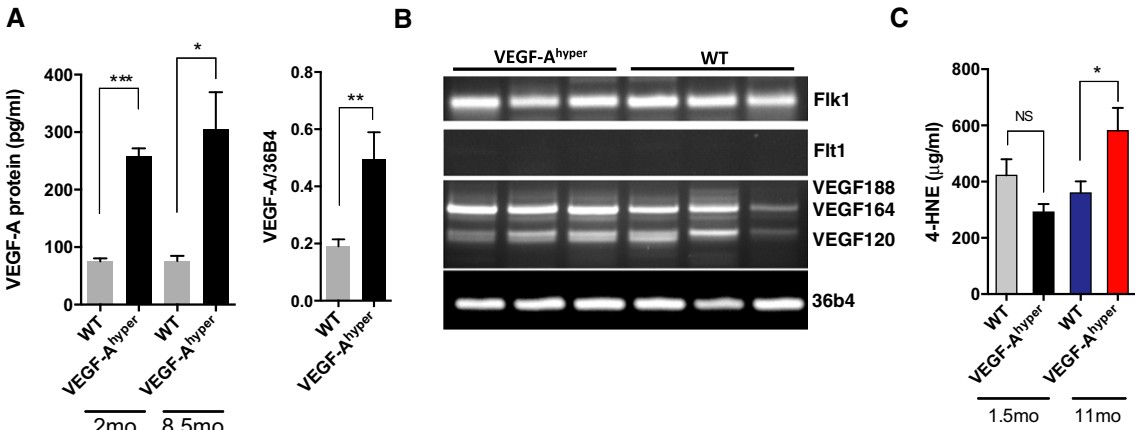

**Figure 2.   Increased VEGF-A expression and 4-HNE levels in aged VEGF-A<sup>hyper</sup> mice.**

A   Left graph: VEGF-A protein levels in lenses of VEGF-A[hyper] mice are increased ~threefold throughout life (measured at 2 and at 8.5 months of age; *n* = 7 mice/group, two independent experiments). Graph shows mean ± SEM. *P-value: 0.0228; ***P-value: 0.0003. Right graph: VEGF-A mRNA levels are similarly increased in lenses of adult VEGF-A[hyper] mice (9 months old; *n* = 7 mice/group, three independent experiments). Graph shows mean ± SEM. **P-value: 0.0095.

B   VEGF-A isoforms (mainly VEGF-A[120] and VEGF-A[164]) and the VEGF-A receptor Flk1 are expressed in the adult lens. Flt1 expression was not detected in the lens. *N* = 3 mice/group.

C   Increased serum levels of the lipid peroxidation marker 4-HNE (a marker of increased oxidative damage) in aged, but not in young VEGF-A[hyper] mice (*P-value: 0.0325; *n* = 7 mice/group, two independent experiments). Graph shows mean ± SEM.

Source data are available online for this figure.

We found that serum levels of the lipid peroxidation marker 4-hydroxynonenal (4-HNE) were significantly increased in aged (11-month-old) VEGF-A[hyper] mice, but not in young (6-week-old) VEGF-A[hyper] mice, when compared with age- and gender-matched control littermate mice, suggesting an age-dependent increase in oxidative stress in these mice as a consequence of increased VEGF-A levels (Fig 2C).

**Higher VEGF-A levels lead to increased oxidative stress markers and ERK hyperactivation in the aging lens and promote age-related nuclear cataract formation that can be inhibited by targeting Nlrp3 or Il1r1**

The co-expression of VEGF-A and its receptor Flk1 in the lens suggests an autocrine/paracrine effect of VEGF-A on lenticular cells. Consistent with this hypothesis, we observed hyperactivation of the VEGF-A downstream target ERK in lenses of VEGF-A[hyper] mice (Fig 3A), which has been implicated in cataractogenesis (Gong *et al*, 2001; Zatechka & Lou, 2002). Importantly, VEGF-A[hyper] mice developed in an age-dependent progressive manner cataracts whose age-related onset and increase in severity resembled aspects of human age-related cataracts (Fig 3B–G), which represent the most frequent form of cataracts and the most common cause of reversible blindness (Rao *et al*, 2011; Petrash, 2013).

In lenses of VEGF-A[hyper] mice with mature cataracts, we observed extensive degeneration of lenticular cells with vacuolization in the lens cortex (Fig 3D and E). Similar histologic changes are also observed in human senile cataracts (Shaikh & Janjua, 1997).

Cataract severity was graded according to the extent of morphological opacification of the lens *in vivo*. Mild opacification of lens that still allowed a proper fundus examination was classified as grade 1 (and a completely translucent lens without cataracts was graded as 0), while increased opacification that restricted a full fundus examination was classified as grade 2. Complete opacification of the lens with leukocoria (white pupil) and no red reflex visible from the fundus represented fully matured cataracts and were classified as grade 3 cataracts.

In 9- to 12-month-old VEGF-A[hyper] mice, about 41% of these mice had formed mature (grade 3) cataracts, while no wild-type littermate mice in this age group formed mature cataracts (Fig 4A and Appendix Fig S1). In > 18-month-old VEGF-A[hyper] mice, about 59% of these mice had formed mature (grade 3) cataracts with complete opacification and leukocoria, while only about 12% of wild-type mice in this age group formed mature cataracts (Fig 4A and Appendix Fig S1). In contrast, no mature cataracts were found in young adult VEGF-A[hyper] mice (no mature cataract was observed in 50 VEGF-A[hyper] mice that were up to 3 months old) (Fig 4A and Appendix Fig S1). This progressive age-dependent increase in cataract frequency and severity in VEGF-A[hyper] mice mirrors the clinical course of senile cataract formation in elderly patients.

Targeting Nlrp3 delayed cataract formation in aged mice with normal VEGF-A levels as well as in VEGF-A[hyper] mice (Fig 4B and Appendix Fig S1). Only about 14% of 9- to 12-month-old VEGF-A[hyper]/Nlrp3[−/−] mice formed mature cataracts (grade 3), in contrast to the about 41% of VEGF-A[hyper] mice that formed mature cataracts in this age group (while no mature cataracts were observed in 9- to 12-month-old wild-type or Nlrp3[−/−] mice) (Fig 4B and Appendix Fig S1). Similarly, genetic inactivation of Il1r1 inhibited age-related cataract formation in mice with normal VEGF-A levels as well as in VEGF-A[hyper] mice with increased VEGF-A levels in the lens (Appendix Fig S1). These findings reveal contributory pathogenic roles of VEGF-A and NLRP3 inflammasome-mediated

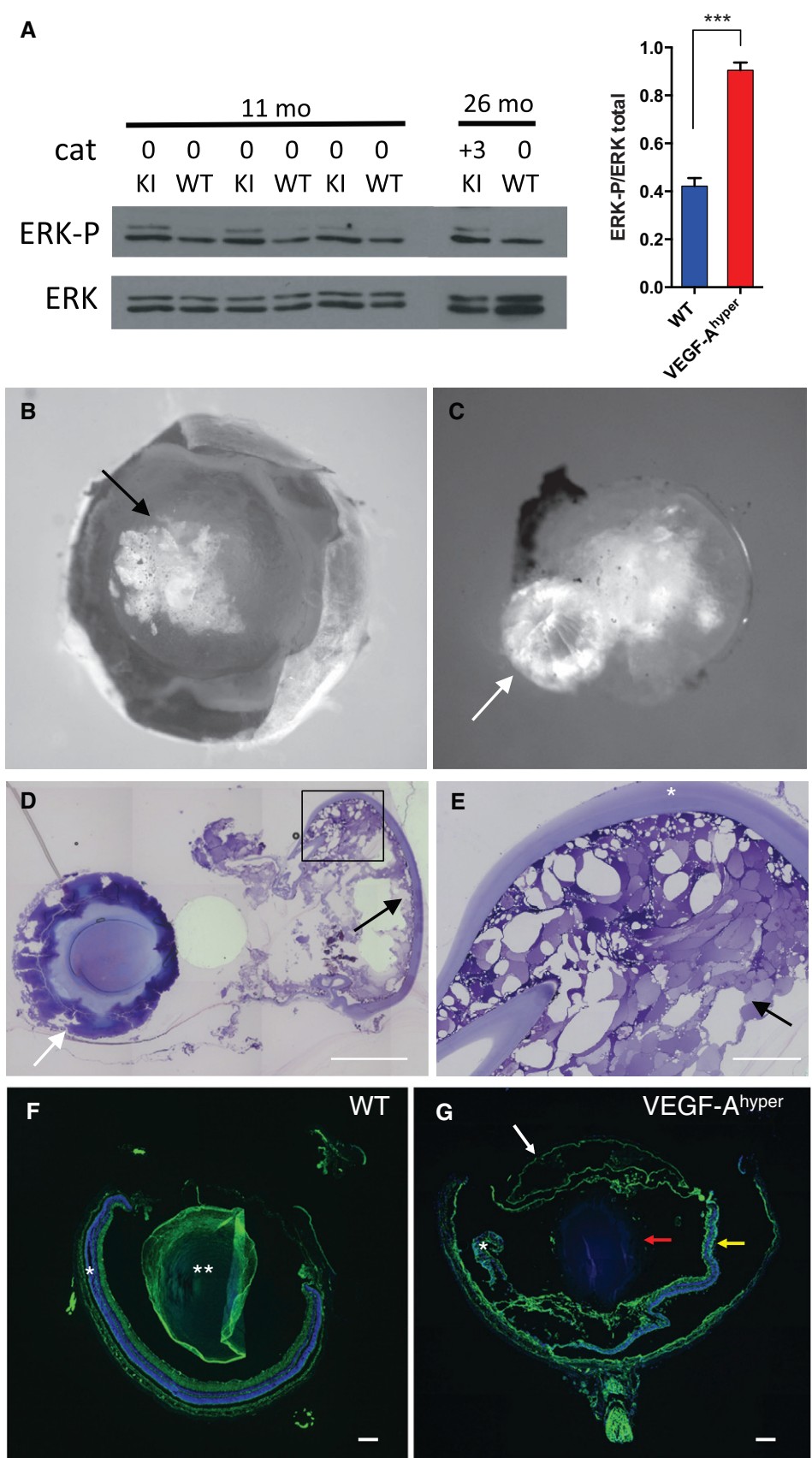

**Figure 3.**

**Figure 3.  Higher VEGF-A levels in the lens cause ERK pathway hyperactivation and age-dependent progressive nuclear cataracts.**

A      Increased ERK1/2 phosphorylation is observed in lenses from VEGF-A[hyper] mice (KI) with increased VEGF-A levels already prior to cataract formation (11-month-old mice with no cataracts (0 cataract)), which is maintained in lenses with mature cataracts (lens of a 26-month-old VEGF-A[hyper] mouse with +3 cataract). Quantification of ERK1/2 phosphorylation in 11-month-old lenses (*n* = 3/group). Graph shows mean ± SEM. ***P-value: 0.0005.

B, C   Cataracts with opacification of the lens (B, black arrow) and extruded nuclei (C, white arrow) are observed with progressive age in VEGF-A[hyper] mice (here a representative 30-month-old lens with a cataract is shown).

D, E   Histology of lens from (B, C) shows extrusion of sclerotic nucleus (white arrows in C and D) and massive degenerative changes in the lens cortex with vacuolization (black arrows in D and E). Scale bars, 500 μm (D) and 100 μm (E).

F, G   Representative histological images of 21-month-old eyes from WT and VEGF-A[hyper] mice. The sclerotic nucleus (red arrow in G) was extruded and the lens cortex and capsule are separated from the nucleus (white arrow in G). The retina in aged VEGF-A[hyper] mice is atrophic and shows degenerative changes (yellow arrow in G), while the retina (*) and lens (**) appear normal in age-matched WT littermate mice. Scale bars, 200 μm. DAPI (blue, nuclei); phalloidin (green).

Source data are available online for this figure.

IL-1β activation for the manifestation of age-related lens opacifications VEGF-A[hyper] mice.

Consistent with these findings, we found increased IL-1β expression in aged lenses with VEGF-A-induced cataracts (Fig 4C). Notably, we have previously observed that increased oxidative stress promotes the secretion of active IL-1β by lens epithelial cells (Marneros, 2013). Thus, our findings suggest that VEGF-A-induced oxidative stress leads to NLRP3 inflammasome-mediated IL-1β activation in the lens that promotes cataractogenesis. As VEGF-A can induce ROS in cells through the activation of NADPH oxidase that lead to autophosphorylation of Flk1 and downstream activation of ERK (Abid *et al*, 2000; Colavitti *et al*, 2002; Monaghan-Benson & Burridge, 2009), and as both ERK hyperactivation and oxidative stress are known inducers of cataracts, we assessed markers of oxidative stress in lenses of VEGF-A[hyper] mice. We found that increased lenticular VEGF-A levels correlated with markers of oxidative stress in lenses of aged VEGF-A[hyper] mice, such as decreased levels of the major antioxidant in the lens, reduced glutathione (already prior to cataract formation), and overexpression of NADPH oxidase gp91[phox] in lenses of aged VEGF-A[hyper] mice with cataracts (Fig 4D and E). Moreover, we found compensatory overexpression of antioxidant enzymes (SOD1, GPx-1, and catalase) in aged lenses with cataracts in VEGF-A[hyper] mice, but not in young lenses without cataracts (Fig 4F).

Thus, our data suggest a previously unknown pathogenic role of increased VEGF-A for cataract formation, which may promote cataractogenesis by increasing oxidative damage and ERK hyperactivation in the lens. VEGF-A-induced oxidative stress may stimulate NLRP3 inflammasome activation and lead to the secretion of active IL-1β to promote cataract formation, consistent with our observation that (i) oxidative stress can induce IL-1β activation in lens epithelial cells *in vitro* (Marneros, 2013), that (ii) increased VEGF-A-induced oxidative damage in lenses with cataracts in VEGF-A[hyper] mice is associated with increased IL-1β expression, and that (iii) genetic inactivation of either Nlrp3 or Il1r1 inhibits cataract formation in VEGF-A[hyper] mice.

**Increased VEGF-A expression specifically in the RPE leads to RPE barrier breakdown via Flk1 signaling and is sufficient for the development of neovascular AMD-like pathologies**

We observed in eyes of all VEGF-A[hyper] mice examined (> 400 mice were examined in total, between ages 6 weeks to 34 months) an age-dependent manifestation of AMD-like pathologies with CNV and progressive RPE/photoreceptor degeneration (Fig 5, Appendix Figs S2 and S3) (Marneros, 2013; Ablonczy *et al*, 2014).

With progressive age, sub-RPE basal laminar-like deposits formed that were associated with the loss of pigment granules in RPE cells, atrophy of RPE cells, and photoreceptor degeneration (Appendix Figs S2 and S3). Thus, AMD-like pathologies occur in 100% of VEGF-A[hyper] mice, while similar pathologies were not observed in any wild-type littermate mice. Moreover, CNV lesions occur with early onset and can be quantitated in VEGF-A[hyper] mice already at 6 weeks of age.

Increased VEGF-A levels in the RPE/retina of VEGF-A[hyper] mice were associated with RPE barrier breakdown, which preceded the manifestation of neovessel growth from the underlying choroidal vessels (Marneros, 2013). We observed in detailed histological analyses and choroidal flat mount stainings that the origin of CNV was the underlying choroidal vasculature and that CNV lesions were separated by RPE cells from the photoreceptors (Fig 5A–C). Similarly as in human AMD, we also found basal laminar deposits and autofluorescent sub-RPE deposit in eyes of aged VEGF-A[hyper] mice, while these were not observed to a comparable extent in eyes of age-matched littermate control mice (Fig 5, Appendix Figs S2 and S3) (Marneros, 2013). Thus, spontaneous CNV lesions form in VEGF-A[hyper] mice in a progressive age-dependent manner that morphologically resemble neovascular AMD lesions in patients, in which neovessels originate from the choroidal vasculature and often protrude into the space between Bruch's membrane and the RPE (Grossniklaus & Green, 1998). These morphological similarities of CNV lesions in VEGF-A[hyper] mice with those in patients with neovascular AMD further demonstrate that these mice serve as a pathophysiologically relevant animal model for AMD.

While we found the RPE to be a major source of VEGF-A expression in the posterior eye, VEGF-A was also expressed in other retinal cells (Fig 1B). Importantly, we observed that activated retinal glia cells migrated toward the RPE at sites of CNV lesion formation and that these glia cells strongly expressed VEGF-A (Marneros, 2013). These findings raise the important question whether increased VEGF-A expression in the RPE is sufficient to induce RPE barrier breakdown and subsequent CNV lesion formation, or whether neovessel growth is a consequence of increased expression of VEGF-A in other cells of the retina, such as in activated retinal glia cells.

Thus, we tested here whether increased VEGF-A expression specifically in the RPE (and not in the retina or circulation) is sufficient to induce RPE barrier breakdown and neovascular AMD-like pathologies, by generating mice that overexpress VEGF-A only in the RPE postnatally (Vmd2Cre[+/WT]/ROSA-STOP[fl/fl]-VEGF-A[164] mice). In Vmd2-Cre[+/WT] mice, Cre recombinase is expressed specifically in the RPE after day 10 postnatally and not in any other retinal

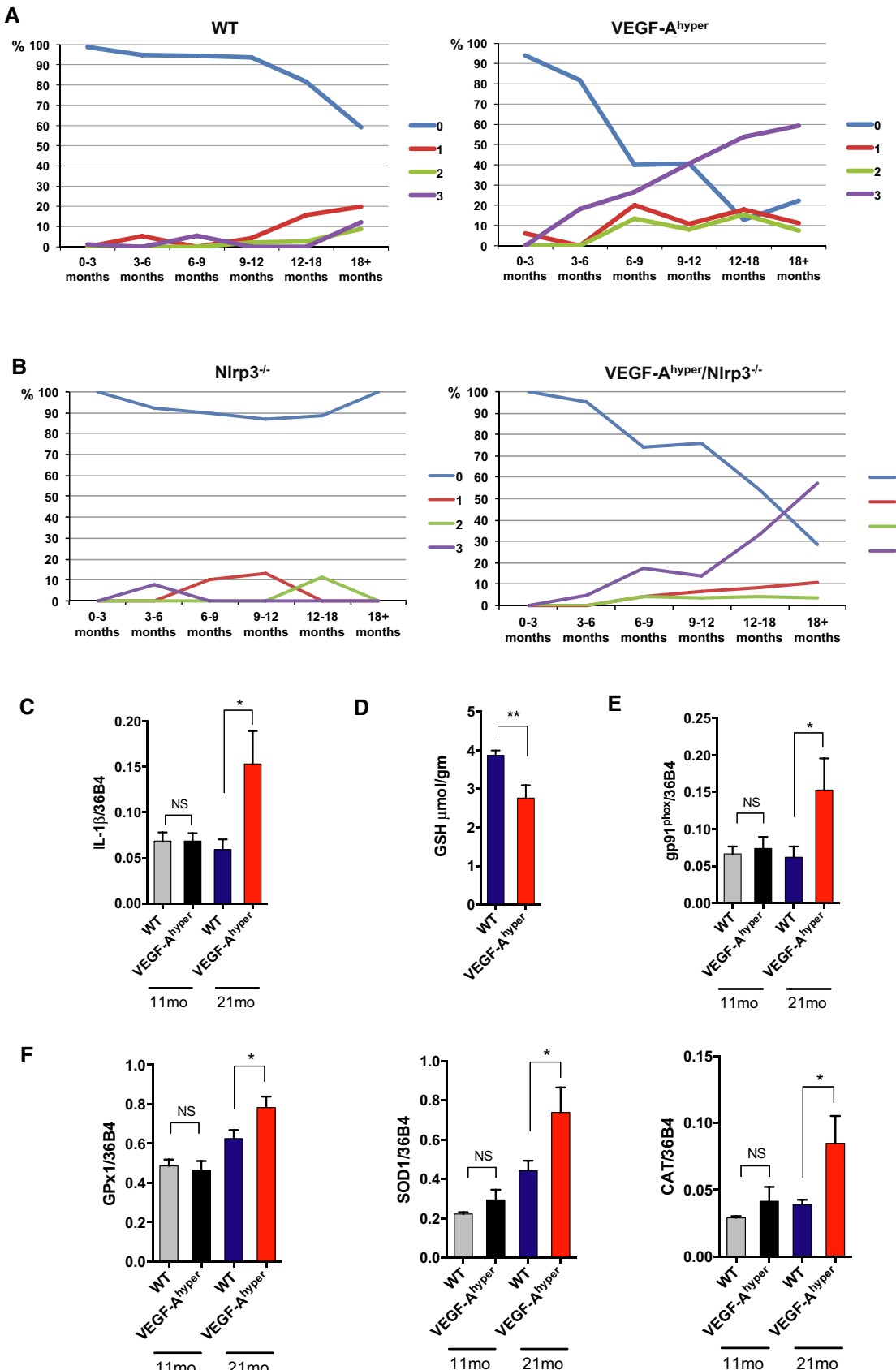

**Figure 4.**

**Figure 4.  Increased lenticular VEGF-A increases markers of oxidative stress in the lens and targeting NLRP3 delays cataract formation.**

A     Cataracts progress with age in VEGF-A^hyper mice. Cataracts were graded from +1 (mild), +2 (moderate) to +3 (mature cataract with fully opacified lens). While only a small subset of > 18-month-old WT mice have mature (+3) cataracts, the majority of VEGF-A^hyper mice of this age group have mature cataracts. Percentile (%) of mice with graded cataracts is indicated in each age group. 0 = no cataract. Absolute mouse numbers of each group are shown in Appendix Fig S1.

B     Targeting Nlrp3 inhibits normal age-dependent cataract formation and delays (but does not prevent) VEGF-A-induced cataract formation in VEGF-A^hyper mice. Percentile (%) of mice with graded cataracts is indicated in each age group. 0 = no cataract. Absolute mouse numbers of each group are shown in Appendix Fig S1.

C     Expression of IL-1β in lenses of 21-month-old VEGF-A^hyper mice with +3 cataracts is increased compared to lenses from age-matched WT mice with +3 cataracts, but not in lenses from 11-month-old mouse groups with no cataracts. *P-value: 0.0259. N = 7/group (three independent experiments).

D     VEGF-A^hyper mice have decreased levels of the major antioxidant in the lens, reduced glutathione (in μmol/gm lens weight), prior to cataract formation. Lenses with 0 to +1 cataracts from 11-month-old mice were used for measurements. **P-value: 0.0097. N = 7/group (two independent experiments).

E, F   Overexpression of NADPH oxidase gp91^phox (*P-value: 0.0435) (E) and of the antioxidant enzymes SOD1 (*P-value: 0.0330), catalase (CAT) (*P-value: 0.0448), and GPx-1 (*P-value: 0.0443) (F) in 21-month-old lenses of VEGF-A^hyper mice with cataracts (+3), but not in 11-month-old lenses with no cataracts. N = 7/group (three independent experiments).

Data information: Graphs show mean ± SEM.

cells (Iacovelli *et al*, 2011). Therefore, in Vmd2Cre^+/WT/ROSA-STOP^fl/fl-VEGF-A^164 mice, the main isoform of VEGF-A in the RPE (VEGF-A^164) is specifically overexpressed in the RPE after day 10 postnatally, which allows us to determine the effects of increased VEGF-A expression in the RPE in the adult without affecting retinal development. Moreover, no Cre-induced abnormalities were observed in young Cre heterozygous mice of this strain that we used for these experiments [we found Cre-induced abnormalities in aged Vmd2-Cre^+/WT mice and in mice homozygous for Cre (He *et al*, 2014)].

Indeed, we observed RPE barrier breakdown and CNV lesions of similar size as in VEGF-A^hyper mice in Vmd2Cre^+/WT/ROSA-STOP^fl/fl-VEGF-A^164 mice, confirming that increased VEGF-A expression specifically in the RPE is sufficient to induce CNV (Fig 6A–C). As observed in CNV lesions of VEGF-A^hyper mice and in human neovascular AMD lesions, CNV lesions originated from the underlying choroidal vessels and were covered by RPE cells also in Vmd2Cre^+/WT/ROSA-STOP^fl/fl-VEGF-A^164 mice (Fig 6A). These findings suggest that RPE-derived VEGF-A induces RPE barrier breakdown and that this is a critical pathogenic step that is required for CNV lesion formation.

VEGF-A-induced RPE barrier breakdown *in vitro* is mediated by signaling through the VEGF-A receptor Flk1 (Ablonczy & Crosson, 2007). Thus, we tested whether inactivation of Flk1 specifically in the RPE would inhibit VEGF-A-induced RPE barrier breakdown and subsequent CNV lesion formation in VEGF-A^hyper mice, by generating Vmd2Cre^+/WTFlk1^fl/flVEGF-A^hyper mice. In these mice, staining of choroidal flat mounts reveals nuclear co-localization of β-gal expression (reflecting increased VEGF-A expression) and Cre recombinase (reflecting Flk1 inactivation). Thus, these mice have increased VEGF-A expression in the RPE, while lacking the Flk1 receptor in the same RPE cells. We found that Vmd2Cre^+/WT Flk1^fl/flVEGF-A^hyper mice (in which the majority of RPE cells were Cre^+) showed no RPE barrier breakdown and CNV lesions even at an advanced age (*n* = 15 mice, up to 8-month-old mice were examined), when VEGF-A^hyper mice have developed extensive large CNV lesions (Fig 6D and E, and Appendix Fig S4). Notably, both Vmd2Cre^+/WTFlk1^fl/fl mice and Vmd2Cre^+/WTFlk1^fl/fl VEGF-A^hyper mice had no significant RPE abnormalities on choroidal flat mounts.

These findings suggest that VEGF-A-induced RPE barrier breakdown occurs in a Flk1-dependent manner *in vivo* and is required for CNV lesion formation. Therefore, targeting Flk1 signaling

in the RPE may prevent the development of neovascular AMD-like pathologies, thereby providing a novel therapeutic rationale for the prevention of neovascular AMD. Moreover, these observations also validate that the AMD-like pathologies that we have observed in VEGF-A^hyper mice occur indeed due to increased VEGF-A levels in the RPE (and are not due to other strain-specific effects). Thus, our data show in two independent genetic mouse models that increased VEGF-A in the RPE is sufficient to cause CNV lesions that originate from the underlying choroidal vasculature as observed in neovascular AMD, thereby providing strong evidence that an increase in VEGF-A in the RPE is a critical pathogenic contributor to neovascular AMD.

## VEGF-A-induced CNV is promoted by NLRP3 inflammasome-mediated IL-1β activation

In contrast to acute laser injury models of neovascular AMD (He & Marneros, 2013), VEGF-A^hyper mice allow us to investigate which molecular mechanisms do not only contribute to CNV lesion growth (measuring CNV lesion size), but also contribute to their spontaneous age-dependent induction without experimental injury (measuring CNV lesion numbers).

We found increased NLRP3 immunolabeling in the RPE at sites of CNV lesion formation and increased expression of NLRP3 and of proangiogenic IL-1β that is activated by the NLRP3 inflammasome in the RPE/choroids of VEGF-A^hyper mice, while IL-18 expression was not increased (Fig 7A–C). This is consistent with our observation that NLRP3 inflammasome activation (with generation of the active caspase-1 products p10 and p20) occurs in the eyes of these mice at sites of RPE barrier breakdown (Marneros, 2013).

Oxidative stress as well as sublytic complement C5b-9 attack on cells can result in IL-1β release through activation of the NLRP3 inflammasome (Laudisi *et al*, 2013; Triantafilou *et al*, 2013), and C5b-9 was shown to be increased in CNV lesions in human eyes with AMD (Johnson *et al*, 2000; Mullins *et al*, 2014). Furthermore, complement C1q has been shown to be a potent inducer of the NLRP3 inflammasome and to be increased in eyes with AMD as well (Doyle *et al*, 2012). Consistent with these findings, we found NLRP3 inflammasome activation in the RPE of CNV lesions in VEGF-A^hyper mice and accumulation of complement C1q and C5b-9 in their CNV lesions, suggesting a role of both in the activation of the NLRP3 inflammasome in CNV lesions of VEGF-A^hyper mice (Fig 7D–F)

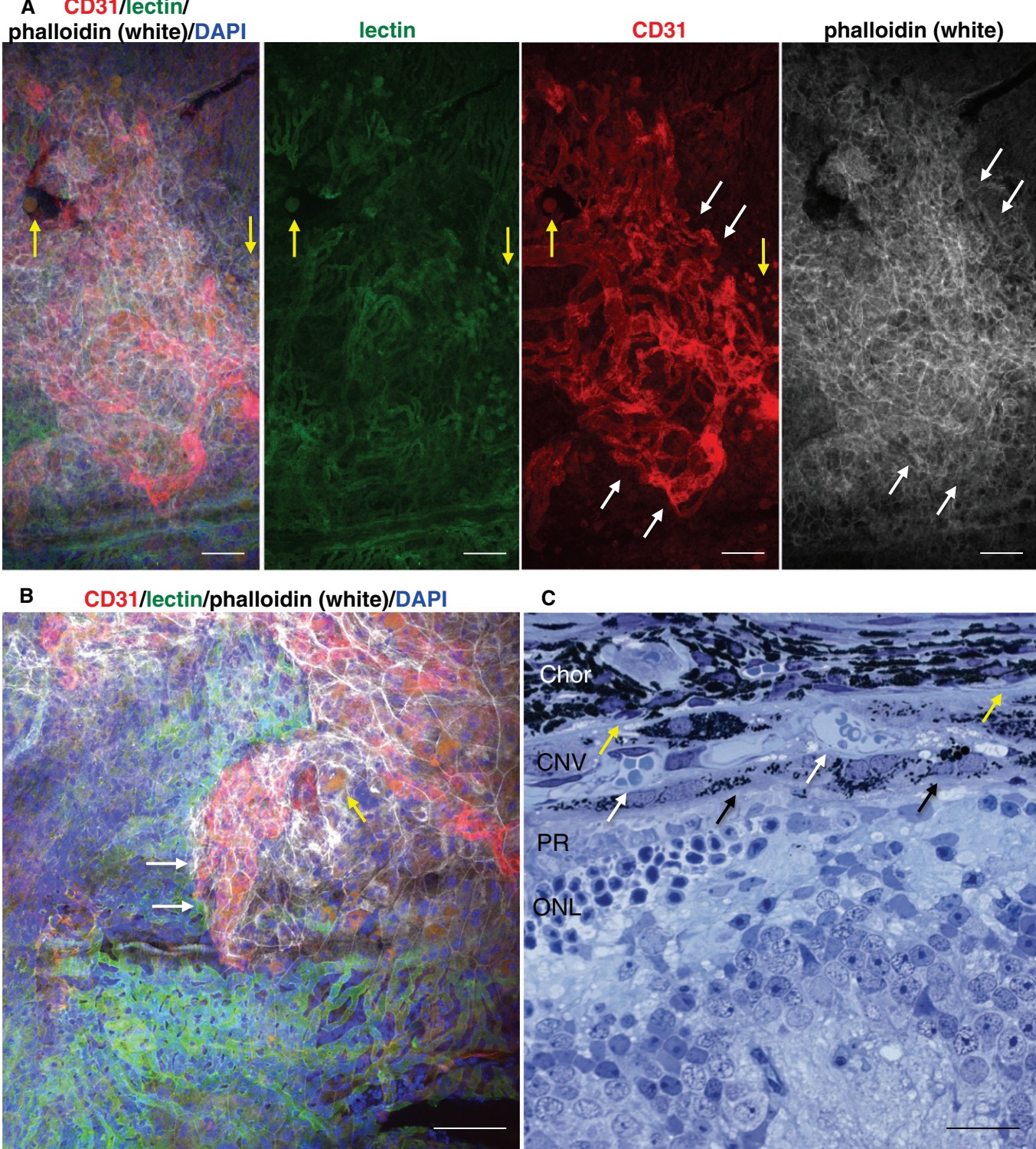

**Figure 5.  Choroidal neovessels originate from the choroidal vasculature in 21-month-old VEGF-A$^{hyper}$ mice.**

A, B    Choroidal flat mount staining of a white VEGF-A$^{hyper}$ mouse shows choroidal vessels [fluorescein-conjugated isolectin B4 (green)] from which neovessels [CD31 (strong red)] originate and that are covered by a RPE cell layer [phalloidin (white)], which separates the CNV lesion from the photoreceptors. White arrows demarcate CNV lesion. Yellow arrows show round autofluorescent deposits. Scale bars, 100 μm.

C    Section through a CNV lesion in a VEGF-A$^{hyper}$ mouse. Bruch's membrane location is indicated by yellow arrows. Choroidal neovessels (white arrows) are covered by a layer of RPE cells (black arrows). Chor: choroid; ONL: outer nuclear layer; PR: photoreceptor outer and inner segments. Scale bar, 20 μm.

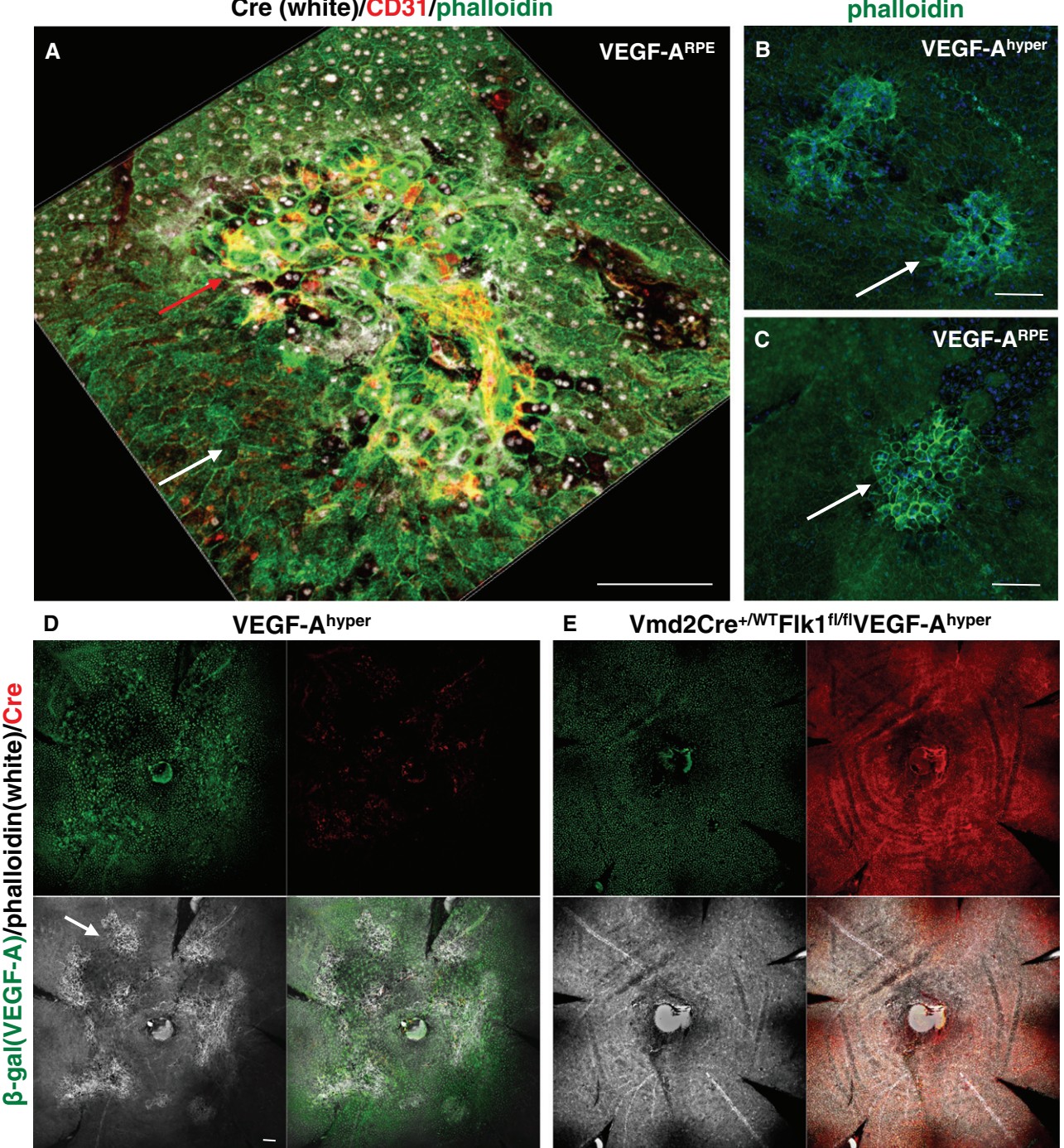

**Figure 6. Increased VEGF-A in the RPE causes neovascular AMD-like pathologies by inducing RPE barrier breakdown through activation of Flk1 signaling in the RPE.**

A–C RPE-specific overexpression of VEGF-A in Vmd2Cre[+/WT]ROSA-STOP[fl/fl]VEGF-A[164] mice (VEGF-A[RPE] mice) results in CNV at sites of VEGF-A overexpression [Cre[+] patches (white nuclei); red arrow], while Cre[−] areas with normal VEGF-A expression show no CNV (white arrow). Neovessels are CD31[+] (red). VEGF-A-induced CNV lesions and RPE barrier breakdown in Vmd2Cre[+/WT]ROSA-STOP[fl/fl]VEGF-A[164] mice resemble those seen in VEGF-A[hyper] mice (B and C, arrows; phalloidin staining in green). Scale bars, 100 µm.

D, E Genetic inactivation of Flk1 specifically in the RPE in VEGF-A[hyper] mice (Vmd2Cre[+/WT]Flk1[fl/fl]VEGF-A[hyper] mice) inhibits RPE barrier breakdown and CNV lesion formation. Choroidal flat mounts of 5-month-old VEGF-A[hyper] mice and Vmd2Cre[+/WT]Flk1[fl/fl]VEGF-A[hyper] mice are shown. β-gal staining shows uniform VEGF-A expression in these mice (green). Cre staining (red) shows co-localization of β-gal and Cre staining in the nuclei of RPE cells in Vmd2Cre[+/WT]Flk1[fl/fl]VEGF-A[hyper] mice (some red areas of staining occur in CNV lesions of VEGF-A[hyper] mice due to labeling of mouse IgG and not Cre, as anti-mouse IgG secondary antibodies were used to detect the mouse anti-Cre antibody). Phalloidin staining (white) reveals extensive RPE barrier breakdown and CNV lesions (arrow) in VEGF-A[hyper] mice, which is not observed in Vmd2Cre[+/WT]Flk1[fl/fl]VEGF-A[hyper] mice. Scale bar, 100 µm.

(Marneros, 2013). The observation of key markers that accumulate in human eyes with AMD also in evolving AMD-like lesions in eyes of VEGF-A$^{hyper}$ mice (e.g. the accumulation of complement C1q and C5b-9) further supports the clinical significance and pathophysiological relevance of findings in this AMD mouse model for the human disease.

The NLRP3 inflammasome activates IL-1β and IL-18, and consistent with the increased expression of NLRP3 and IL-1β in the RPE/choroids of eyes of VEGF-A$^{hyper}$ mice, we found that the NLRP3 inflammasome promotes VEGF-A-induced CNV lesion formation through the proangiogenic factor IL-1β, as genetic inactivation of Nlrp3 or Il1r1 potently reduced the numbers of CNV lesions that formed in VEGF-A$^{hyper}$ mice (Fig 8A–E and Appendix Fig S5A and B). We counted CNV lesions and measured their circumferential area in choroidal flat mount stainings in 6-week-old mice of each experimental group. VEGF-A$^{hyper}$ mice at 6 weeks of age had an average of 18.41 CNV lesions per eye, while VEGF-A$^{hyper}$/Il1r1$^{-/-}$ mice (lacking IL-1β signaling activity) had an average of only 8.79 CNV lesions per eye (P-value compared to VEGF-A$^{hyper}$ mice of 0.0085). VEGF-A$^{hyper}$/Nlrp3$^{-/-}$ mice (lacking NLRP3 inflammasome activity that activates IL-1β) had only 6.26 CNV lesions per eye (P-value compared to VEGF-A$^{hyper}$ mice of 0.0002). No CNV lesions were observed in age-matched littermate control mice (obtained from heterozygous matings between VEGF-A$^{hyper}$ mice), or in mice lacking Nlrp3 or Il1r1 (> 30 mice for each of these control groups were examined).

In our initial observations with a small number of mice, we had observed that VEGF-A$^{hyper}$ mice lacking Il18 showed a trend toward having more CNV lesions, but the observed high variability within this group of mice and the small sample number resulted in statistically inconclusive results (Marneros, 2013). Here, we analyzed CNV lesion numbers and sizes in a large number of age-matched (6-week-old) mice and could show that targeting Nlrp3 or Il1r1 significantly reduced CNV lesion numbers (but not CNV lesion sizes), while targeting Il18 did not show a statistically significant change in either CNV lesion numbers or their size (Fig 8A and B, and Appendix Fig S5A and B). Thus, VEGF-A-induced CNV lesion induction in this mouse model of AMD is promoted by NLRP3 inflammasome-mediated generation of the highly proangiogenic factor IL-1β (but not by IL-18). These findings in the posterior eye resemble our observations in the lenses of aged VEGF-A$^{hyper}$ mice and demonstrate that both VEGF-A-induced age-related cataract formation and CNV are strongly promoted by NLRP3 inflammasome-mediated IL-1β activation, suggesting a shared VEGF-A-induced pathomechanism for these distinct aging eye pathologies.

**Targeting caspase-1/caspase-11 potently inhibits CNV lesion formation and growth**

Caspase-1 activation is essential for inflammasome-mediated generation of active IL-1β and IL-18, while the proinflammatory caspase-11 has a critical role in activating caspase-1 and has also been shown to lead to pyroptotic cell death in a caspase-1-independent manner (Kayagaki et al, 2011). Moreover, caspase-11 promotes cell migration during inflammation and lack of caspase-11 inhibits macrophage migration (Li et al, 2007). We and others have observed that infiltrating macrophages promote laser-induced CNV

formation (Espinosa-Heidmann et al, 2003; Sakurai et al, 2003; Caicedo et al, 2005; Semkova et al, 2011; Shi et al, 2011; He & Marneros, 2013; Marneros, 2013). Thus, we hypothesized that the lack of both inflammatory caspases (caspase-1 and caspase-11) may inhibit CNV formation more potently than targeting Nlrp3 or IL-1β signaling alone. Indeed, we observed that genetic inactivation of caspase-1/caspase-11 in VEGF-A$^{hyper}$ mice strongly inhibited both CNV lesion formation and growth, while targeting Nlrp3 or Il1r1 inhibited CNV lesion formation less potently and had no effect on CNV lesion growth (Fig 8A–E and Appendix Fig S5A and B). Average CNV lesions per eye were for 6-week-old VEGF-A$^{hyper}$ mice 18.41, for VEGF-A$^{hyper}$/Il1r1$^{-/-}$ mice 8.79, for VEGF-A$^{hyper}$/Nlrp3$^{-/-}$ mice 6.26, but for VEGF-A$^{hyper}$/Casp1$^{-/-}$/Casp11$^{-/-}$ mice only 2.71 (P-value compared to VEGF-A$^{hyper}$ mice of 0.0002). Average CNV lesion size in 6-week-old VEGF-A$^{hyper}$ mice was 22,265 μm$^2$, while CNV lesions in VEGF-A$^{hyper}$/Casp1$^{-/-}$/Casp11$^{-/-}$ mice were on average only 10,137 μm$^2$ in size (P-value compared to VEGF-A$^{hyper}$ mice of < 0.0001). These findings reveal caspase-1/caspase-11 as potential novel promising therapeutic targets in neovascular AMD.

**CNV lesion size is regulated by autophagy and TLR2 signaling**

The NLRP3 inflammasome can be activated by oxidative stress and TLR signaling and inhibited by autophagy in vitro (Netea et al, 2009; Into et al, 2010; Miura et al, 2013). Notably, we have observed increased expression of TLR2 in RPE cells of VEGF-A$^{hyper}$ mice and strong labeling for TLR2 in CNV lesions of these mice (Fig 7G) (Marneros, 2013). Moreover, TLR2 has been shown to promote angiogenesis in response to oxidative stress (West et al, 2010).

As we have shown that VEGF-A can induce oxidative stress in the RPE in vitro and as the RPE in VEGF-A$^{hyper}$ mice shows evidence of increased oxidative damage as well (Marneros, 2013), we hypothesized that increased oxidative stress-induced TLR2 signaling may promote NLRP3 inflammasome-mediated CNV lesions in VEGF-A$^{hyper}$ mice, while autophagy (having a potential inhibitory effect on the NLRP3 inflammasome) may reduce CNV. Consistent with this hypothesis, we found a strongly reduced CNV lesion size in VEGF-A$^{hyper}$/Tlr2$^{-/-}$ mice and a moderate increase in CNV lesion size in VEGF-A$^{hyper}$ mice with reduced autophagy (due to heterozygosity for the critical autophagy gene beclin-1) (Fig 8A and B and Appendix Fig S5A and B). While the average CNV lesion size in 6-week-old VEGF-A$^{hyper}$ mice was 22,265 μm$^2$, CNV lesions in 6-week-old VEGF-A$^{hyper}$/Tlr2$^{-/-}$ mice were on average only 11,742 μm$^2$ in size (P-value compared to VEGF-A$^{hyper}$ mice of < 0.0001). Notably, CNV lesion numbers were not affected by deficiency for Tlr2. These findings suggest that increased VEGF-A induces oxidative stress and a complement-mediated inflammatory reaction that lead to the activation of the NLRP3 inflammasome, which promotes CNV lesion induction through the generation of active IL-1β, while CNV lesion growth is regulated in part by TLR2 signaling and autophagy. Moreover, our findings show that increased VEGF-A plays a direct pathogenic role in the development of AMD-like pathologies, supporting the observations in human AMD.

Notably, adult Nlrp3$^{-/-}$, Casp1$^{-/-}$/Casp11$^{-/-}$, Il1r1$^{-/-}$, Il18$^{-/-}$, Tlr2$^{-/-}$, Becn1$^{-/+}$, or Atg16l1$^{hypo}$ mice showed no obvious

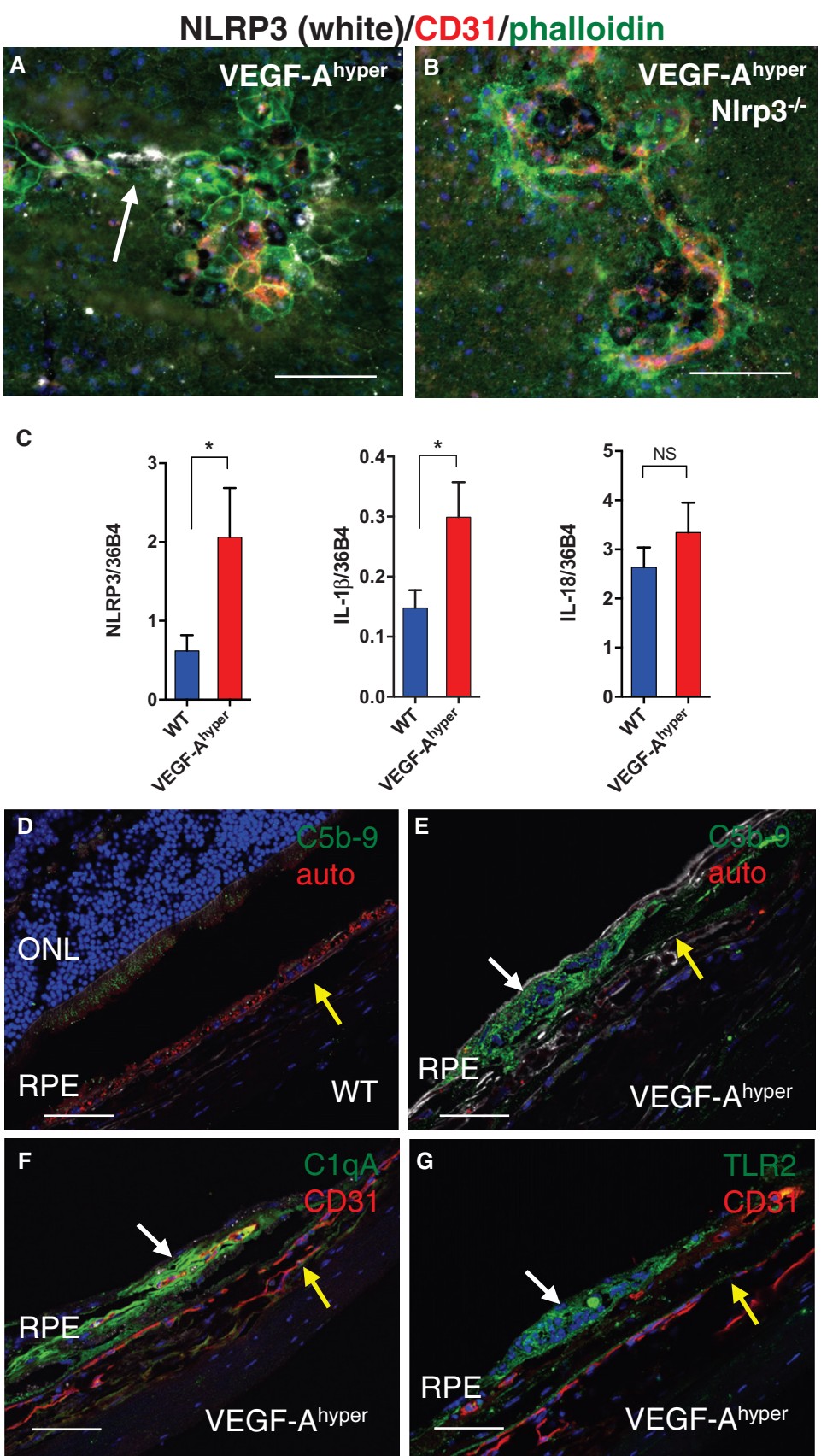

**Figure 7.**

**Figure 7.  Increased NLRP3, complement C1q and C5b-9 in CNV lesions of VEGF-A$^{hyper}$ mice.**

A    NLRP3 is expressed in RPE cells at sites of CNV lesions (arrow; white). Scale bar, 100 μm.
B    Lack of NLRP3 staining in VEGF-A$^{hyper}$/Nlrp3$^{-/-}$ mice (control). Scale bar, 100 μm.
C    Expression of NLRP3 (*P-value: 0.0478) and IL-1β (*P-value: 0.0398), but not of IL-18 (P-value: 0.3529), is increased in RPE/choroid lysates from VEGF-A$^{hyper}$ mice when compared to matched control littermate mice. N = 7 mice/group (three independent experiments). Graphs show mean ± SEM.
D–G  Complement pathway activation (immunolabeling for the product of complement pathway activation C5b-9, an inducer of the NLRP3 inflammasome) is observed in CNV lesions of VEGF-A$^{hyper}$ mice (white arrow in E), but not in age-matched control mice (D). Co-localization of C5b-9 in the same CNV lesion is observed with complement C1qA (the initiator of the classical complement pathway and an inducer of the NLRP3 inflammasome) (F) and with TLR2 (G) (white arrows). Autofluorescence (auto) in red in (D) and (E). CD31$^+$ vessels in red in (F) and (G). Yellow arrows show site of choroid. Scale bars, 50 μm.

abnormalities of the choroidal vasculature and developed no spontaneous CNV lesions (> 30 age-matched mice per group were examined).

As we have observed that genetic inactivation of caspase-1/caspase-11 or Tlr2 results in reduced CNV growth in VEGF-A$^{hyper}$ mice, we next tested whether we can obtain further support for these findings in a non-genetic independent experimental mouse model of CNV and whether pharmacologic inhibition of these pathways may provide a therapeutic approach to inhibit CNV growth. Thus, we tested the effects of treatment of wild-type mice either with a small-molecule inhibitor of caspase-1 or with neutralizing anti-TLR2 antibodies in laser injury-induced CNV experiments. Laser injury to the RPE/choroid induces CNV lesion formation in this model that can be readily quantified 7 days after experimental injury (He & Marneros, 2013). Consistent with the observed reduced CNV lesion growth in VEGF-A$^{hyper}$/Casp1$^{-/-}$/Casp11$^{-/-}$ mice compared to VEGF-A$^{hyper}$ mice, we observed that pharmacologic inhibition of caspase-1 significantly reduced laser-induced CNV lesion growth when compared to DMSO-treated control mice (Fig 8F). Furthermore, treatment with neutralizing TLR2 antibodies also significantly reduced laser-induced CNV lesion growth, similarly as observed when targeting Tlr2 genetically in VEGF-A$^{hyper}$ mice (Fig 8F).

**Progressive VEGF-A-induced RPE and photoreceptor degeneration and decline in visual function are mediated in part by NLRP3 inflammasome-dependent IL-1β activation and can be inhibited by targeting Nlrp3, caspase-1/caspase-11, or Il1r1**

VEGF-A$^{hyper}$ mice develop progressive basal laminar sub-RPE deposits and degeneration of the RPE and photoreceptors, resembling aspects of non-exudative AMD (Fig 9B and Appendix Figs S2 and S3). Initial degenerative changes are observed already in young VEGF-A$^{hyper}$ mice, but these changes progress with increasing age and are particularly prominent in mice > 12 months of age. In addition, autofluorescent sub-RPE deposits are observed in aged VEGF-A$^{hyper}$ mice (Figs 5A and B, and 11B). These age-dependent eye pathologies occurred also at sites devoid of CNV lesion formation (making it unlikely that they are a consequence of CNV) and were not observed in age- and gender-matched wild-type littermate control mice or in VEGF-A$^{hypo}$ mice, which are hypomorphic for VEGF-A and express β-galactosidase from the endogenous VEGF-A locus (thereby excluding a contributory role of β-galactosidase expression for these pathologies) (Fig 9A and I). Similarly as in human AMD, the age-dependent AMD-like pathologies in VEGF-A$^{hyper}$ mice were associated with progressive attenuation of visual function with diminished electroretinogram (ERG) amplitudes

and reduced rhodopsin levels, providing further evidence for the clinicopathologic relevance of these morphologic observations for human AMD (Ablonczy et al, 2014).

In preliminary experiments with 6-week- to 2-month-old VEGF-A$^{hyper}$ mice that lacked either Nlrp3, Il1r1, or Il18, we had previously observed the occasional occurrence of some VEGF-A-induced chorioretinal pathologies, but we had not performed a detailed analysis of the age-dependent AMD-like pathologies in these mice with progressive age (Marneros, 2013). Thus, we investigated here whether inactivation of inflammasome components can inhibit not only VEGF-A-induced CNV, but also degenerative changes of the RPE and photoreceptors in aged VEGF-A$^{hyper}$ mice.

Indeed, genetic inactivation of Casp1/Casp11, Nlrp3, or Il1r1 inhibited the manifestation of age-dependent RPE and photoreceptor degeneration in aged VEGF-A$^{hyper}$ mice, while this effect was not observed in VEGF-A$^{hyper}$ mice that lacked Il18 or that were heterozygous for the key autophagy gene beclin-1 (Fig 9B–H). For example, 21-month-old VEGF-A$^{hyper}$ mice showed severe degenerative and atrophic changes of the RPE, thick basal laminar sub-RPE deposits, shortening of photoreceptor outer and inner segments, and attenuation of the photoreceptor outer nuclear layer (ONL) (Fig 9B and Appendix Figs S2 and S3). In contrast, these pathologies were much reduced in VEGF-A$^{hyper}$ mice that lacked Nlrp3, Casp1/Casp11, or Il1r1 (Fig 9C–E).

Notably, RPE and photoreceptor degeneration was still present in aged VEGF-A$^{hyper}$/Il18$^{-/-}$ mice, which formed thick sub-RPE deposits that protruded into the RPE plane (Fig 9F and G).

In order to obtain a clinicopathological correlation of these morphological abnormalities of the RPE and photoreceptors, and to demonstrate that the visual function decline in aged VEGF-A$^{hyper}$ mice can be rescued by targeting NLRP3 inflammasome components, we performed dark-adapted (to assess mainly rod function) and light-adapted (to assess mainly cone function) electroretinograms (ERGs) in 21-month-old mice of these experimental groups. While in 21-month-old VEGF-A$^{hyper}$ mice ERG amplitudes (a- and b-waves) for both dark-adapted and light-adapted conditions were severely diminished (consistent with the severe RPE and photoreceptor degeneration observed by histologic analysis), targeting caspase-1/caspase-11 or Il1r1 in VEGF-A$^{hyper}$ mice partially normalized ERG amplitudes (Fig 10A–C).

The improvement of visual function in aged VEGF-A$^{hyper}$ mice by blocking inflammasome activation is consistent with the inhibition of morphological degenerative changes observed in these mice. The presence of RPE and photoreceptor degeneration in VEGF-A$^{hyper}$/Il18$^{-/-}$ mice is also supported by the observation that targeting Il18 did overall not improve ERG amplitudes and visual function in aged VEGF-A$^{hyper}$ mice (Fig 10A–C).

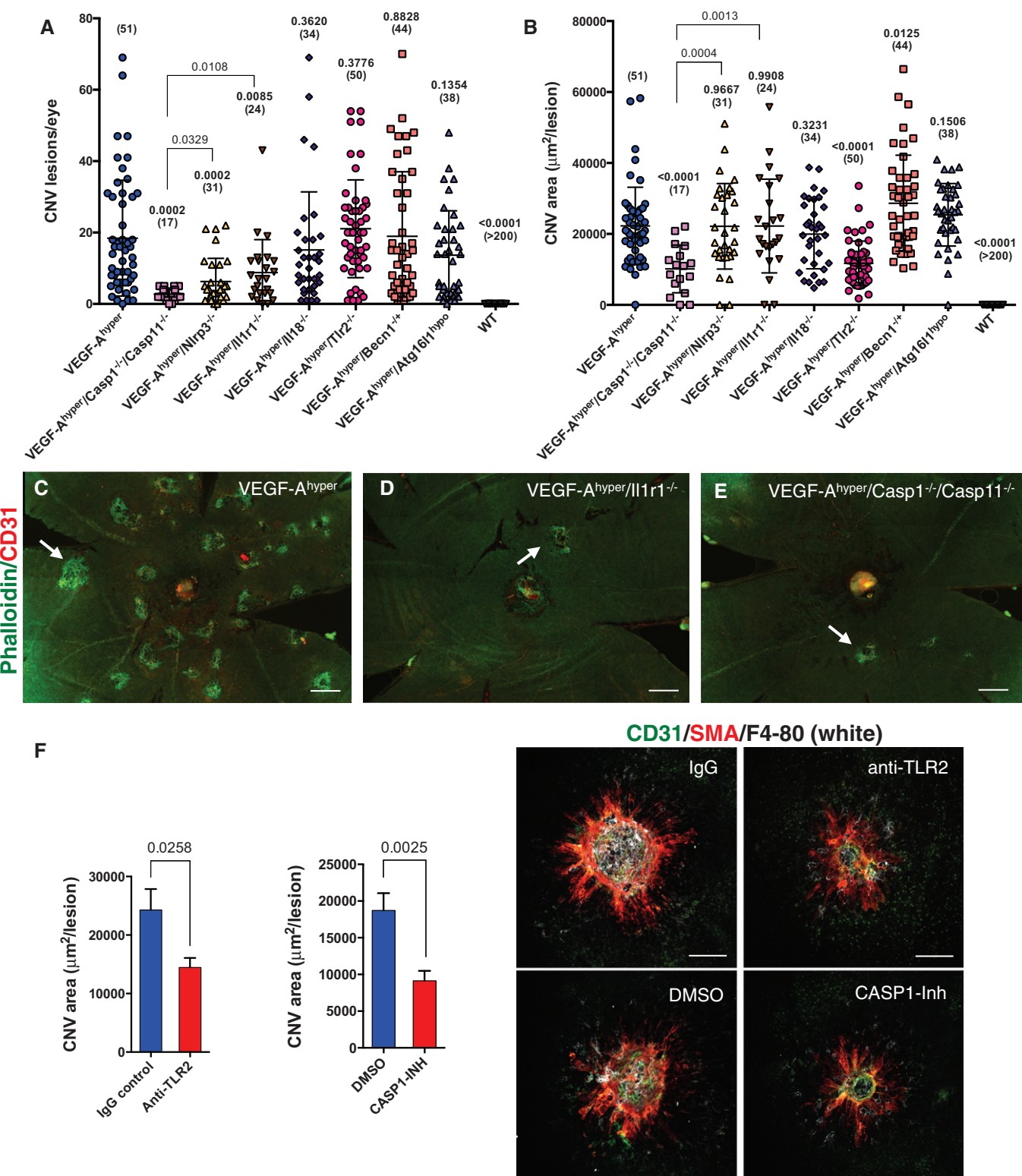

**Figure 8.**

To further strengthen the clinicopathologic correlation of these findings, we assessed additional clinical parameters observed in AMD in those aged mice in which we performed ERGs by fundus imaging, autofluorescence analysis of the fundus, and optical coherence tomography (OCT).

OCTs in aged VEGF-A$^{hyper}$ mice showed severe degenerative changes in the retina and CNV lesions originating from the choroid, which were much reduced in age-matched VEGF-A$^{hyper}$/Casp1$^{-/-}$/Casp11$^{-/-}$ mice and not seen in Casp1$^{-/-}$/Casp11$^{-/-}$ mice (Fig 11A).

Figure 8. VEGF-A-induced AMD-like pathologies occur in an NLRP3 inflammasome- and IL-1β-mediated process, which is regulated by TLR2 signaling and autophagy pathways.

A   CNV lesion numbers per eye in 6-week-old VEGF-A$^{hyper}$ mice compared with VEGF-A$^{hyper}$ mice lacking Casp1/Casp11, Nlrp3, Il1r1, Il18, Tlr2, or those heterozygous for beclin-1 or homozygous for hypomorphic Atg16l1. Casp1/Casp11, Nlrp3, or Il1r1 deficiency results in significantly fewer CNV lesions. Each value represents the total CNV lesion number/eye for an individual mouse. Mean ± SEM is shown. P-values are based on comparison with the VEGF-A$^{hyper}$ mouse group. Absolute numbers of mice per group are indicated in parentheses. P-values were calculated with a two-tailed unpaired Student's t-test. Separately, P-values are shown between VEGF-A$^{hyper}$/Casp1$^{-/-}$/Casp11$^{-/-}$ and VEGF-A$^{hyper}$/Nlrp3$^{-/-}$ or VEGF-A$^{hyper}$/Il1r1$^{-/-}$ mouse groups. WT mice developed no CNV lesions.
B   Average CNV lesion area (in μm$^2$/lesion) in 6-week-old VEGF-A$^{hyper}$ mice compared with VEGF-A$^{hyper}$ mice lacking Casp1/Casp11, Nlrp3, Il1r1, Il18, Tlr2, or those heterozygous for beclin-1 or homozygous for hypomorphic Atg16l1. Lack of Tlr2 or Casp1/Casp11 significantly reduces CNV lesion size, while reduced autophagy moderately increases CNV size. Each value represents the average CNV lesion size/eye for an individual mouse. Mean ± SEM is shown. P-values are based on comparison with the VEGF-A$^{hyper}$ mouse group. Absolute numbers of mice per group are indicated in parentheses. P-values were calculated with a two-tailed unpaired Student's t-test. Separately, P-values are shown between VEGF-A$^{hyper}$/Casp1$^{-/-}$/Casp11$^{-/-}$ and VEGF-A$^{hyper}$/Nlrp3$^{-/-}$ or VEGF-A$^{hyper}$/Il1r1$^{-/-}$ mouse groups.
C–E Representative choroidal flat mount images show multifocal CNV lesions in VEGF-A$^{hyper}$ mice and a significant reduction in CNV numbers in those VEGF-A$^{hyper}$ mice that lack either Casp1/Casp11 or Il1r1 (and therefore IL-1β signaling). Staining for phalloidin in green and for CD31 in red. Scale bars, 200 μm.
F   Laser-induced CNV experiments demonstrate that treatment with a caspase-1 inhibitor (compared to DMSO-treated controls) or a neutralizing TLR2 antibody (compared to isotype-matched IgG-treated controls) inhibits CNV lesion growth. P-values are shown. Mean ± SEM is shown. N = 15 mice/group. Representative CNV lesions in choroidal flat mounts are shown. Scale bars, 100 μm.

Autofluorescence imaging of the fundus showed areas of strong autofluorescence in fundus images of aged VEGF-A$^{hyper}$ mice, a feature associated with autofluorescent sub-RPE deposits as observed in AMD (Fig 11B). Notably, VEGF-A$^{hyper}$/Casp1$^{-/-}$/Casp11$^{-/-}$ mice and Casp1$^{-/-}$/Casp11$^{-/-}$ mice showed much fewer or no autofluorescent deposits.

Consistent with the morphological and functional findings of RPE and photoreceptor degeneration in aged VEGF-A$^{hyper}$ mice, we also found abnormal pigmentation of the fundus in these mice, while VEGF-A$^{hyper}$/Casp1$^{-/-}$/Casp11$^{-/-}$ mice had a largely normal appearing fundus compared to the Casp1$^{-/-}$/Casp11$^{-/-}$ mice (Fig 11C). Thus, the inhibition of the observed AMD-like pathologies in VEGF-A$^{hyper}$ mice by targeting caspase-1/caspase-11 is supported by the functional assessment of visual function (ERGs) and by clinical imaging techniques (OCTs and fundus imaging) in these mice. In conclusion, NLRP3 inflammasome-mediated IL-1β activation promotes VEGF-A-induced progressive eye pathologies that resemble aspects of both neovascular AMD (CNV) and non-exudative AMD (RPE/photoreceptor degeneration and accumulation of sub-RPE deposits).

## Discussion

Our findings suggest a direct pathogenic role of increased VEGF-A for the manifestation of age-related lens opacifications and for both neovascular and non-exudative AMD-like pathologies through NLRP3 inflammasome-dependent mechanisms. Thus, VEGF-A-induced pathways and NLRP3 inflammasome components may serve as novel therapeutic targets to inhibit these major common aging diseases of the eye without impairing the adult microvasculature, as would be expected to occur when targeting VEGF-A continuously.

Importantly, increased VEGF-A levels accelerated age-related changes that normally occur in these tissues and promoted the transition of these normally occurring aging changes to pathologic organ dysfunction. For example, the progressive opacification in the lens that occurs with increasing age in wild-type mice was increased and exacerbated in VEGF-A$^{hyper}$ mice that have increased VEGF-A levels

in their lenses, leading to more frequent and earlier-onset cataracts when compared to age-matched control mice.

Thus, reducing increased VEGF-A activity in aging tissues may inhibit both normal aging-associated functional declines in these tissues and the manifestation of age-related diseases. For example, reducing environmental factors or other changes that promote increased hypoxia or oxidative stress may lead to normalization of VEGF-A levels in aging tissues and inhibit the progression of these aging-associated pathologies.

This hypothesis is consistent with clinical studies suggesting that a diet high in antioxidants may slow the progression of advanced AMD and of age-related cataracts (Chew et al, 2013; Rautiainen et al, 2014).

However, an increase in VEGF-A may also have beneficial effects in some aging tissues, where induced VEGF-A expression may be critical to stimulate compensatory angiogenesis and improve tissue perfusion in response to tissue hypoxia. Thus, targeting VEGF-A directly may have detrimental effects in these conditions.

This is consistent with observed adverse effects of anti-VEGF-A therapies, likely as a consequence of abolishing important roles of VEGF-A for the maintenance and regulation of the adult microvasculature (Kamba et al, 2006). These effects of anti-VEGF-A therapies have been reported to precipitate clinical adverse reactions in some patients, including hypertension, increased risk of hemorrhage or thromboembolic events, or cardiac ischemia (Kamba & McDonald, 2007). Thus, it is highly desirable to identify novel therapeutic approaches that inhibit the pathogenic effects of increased VEGF-A levels for the manifestation of aging diseases without causing the side effects that occur when neutralizing VEGF-A directly. Our findings suggest that inhibition of NLRP3 inflammasome components could serve as such a comprehensive novel therapeutic approach that can inhibit the downstream effects of VEGF-A that promote age-related pathologies, without causing the adverse side effects of anti-VEGF-A therapies and without impairing tissue vascularization.

Our data reveal a previously unknown role of increased VEGF-A levels for the induction of oxidative stress in aged lenses that is associated with the development of age-related lens opacifications, and we show that targeting Nlrp3 or Il1r1 inhibits VEGF-A-induced age-related cataractogenesis.

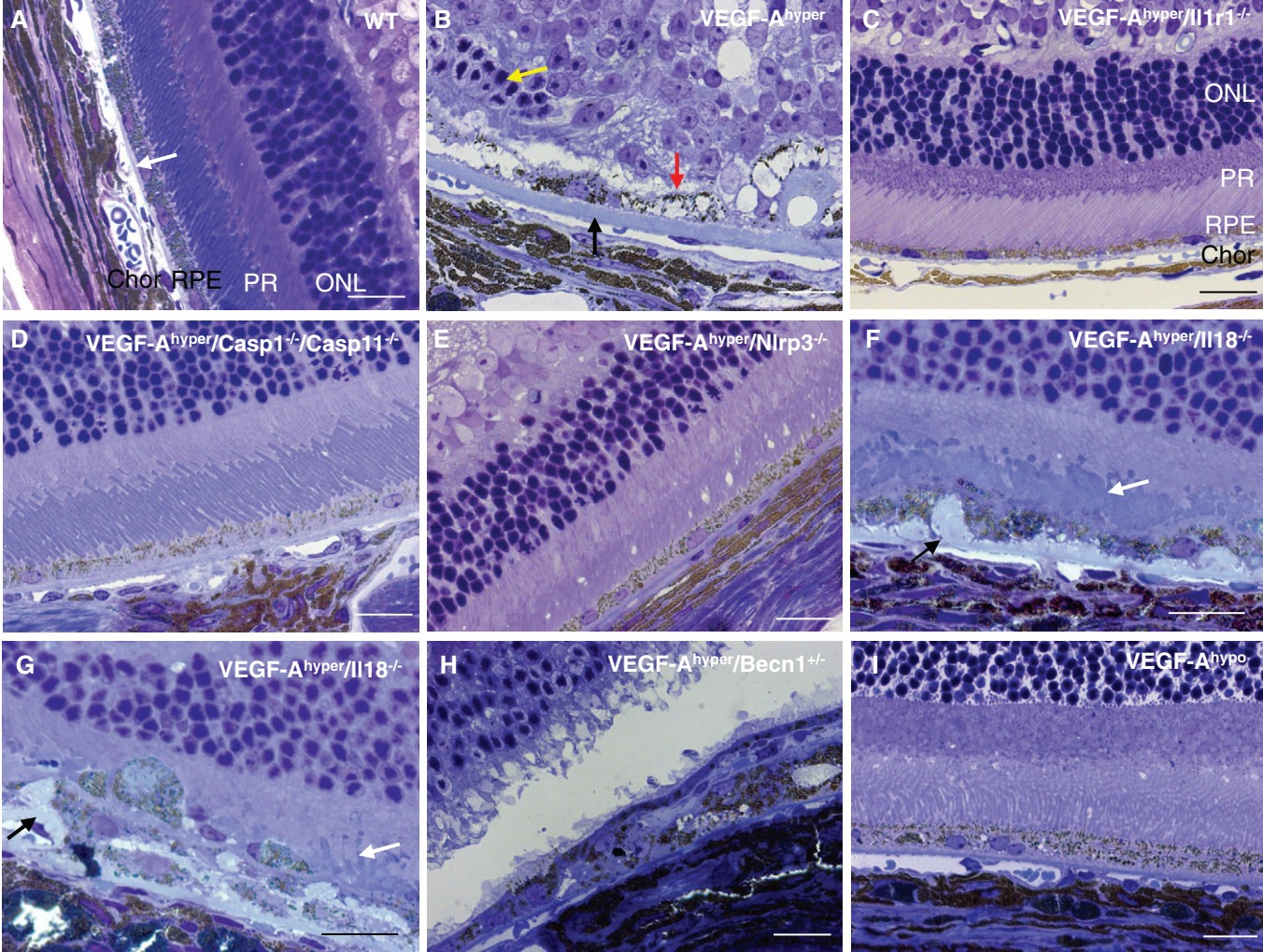

**Figure 9. VEGF-A-induced degeneration of the RPE and photoreceptors can be inhibited by targeting Nlrp3, Il1r1, or caspase-1/caspase-11 in VEGF-A^hyper mice.**

A   No significant abnormalities of the choroid, RPE, or retina are seen in control wild-type littermate mice. Representative image of a 28-month-old control eye is shown. White arrow indicates normal Bruch's membrane. Chor: choroid; RPE: retinal pigment epithelium; PR: photoreceptor outer and inner segments; ONL: outer nuclear layer.

B   Aged VEGF-A^hyper mice develop a progressive age-dependent degeneration of the RPE (red arrow) and the photoreceptors (yellow arrow) with accumulation of thick sub-RPE basal laminar deposits (black arrow). These pathologies occur also at sites devoid of CNV lesions.

C–E Genetic inactivation of Casp1/Casp11, Nlrp3, or Il1r1 strongly inhibits the manifestation of RPE and photoreceptor degeneration in aged VEGF-A^hyper mice.

F, G VEGF-A^hyper/Il18^−/− mice show a significant RPE and photoreceptor degeneration, with shortened and irregular photoreceptor outer segments (white arrows). Thick sub-RPE deposits that protrude into the RPE plane are observed as well (black arrows).

H   VEGF-A^hyper/Becn1^+/− mice show RPE and photoreceptor degeneration.

I   In contrast, age-matched VEGF-A^hypo mice that are hypomorphic for VEGF-A and express β-galactosidase from the endogenous VEGF-A locus do not show these eye pathologies.

Data information: Representative images in (B–I) are from eyes of 21-months old mice. Scale bars, 20 μm.

Similarly, we demonstrate here that VEGF-A-induced CNV and degeneration of the RPE and photoreceptors can be potently inhibited by genetically inactivating NLRP3 inflammasome components or Il1r1 in VEGF-A^hyper mice. Thus, pharmacologic inhibition of the NLRP3 inflammasome or of IL-1β signaling may inhibit both AMD and senile cataract progression. For example, we show here that a caspase-1 inhibitor potently inhibited laser-induced CNV growth (Fig 8F). Similarly, a specific small chemical inhibitor of the

NLRP3 inflammasome has been reported that may thus be used therapeutically to inhibit these aging diseases (Coll *et al*, 2015). Alternatively, IL-1β blockers could also have clinical benefit in the prevention or inhibition of AMD or age-related cataract formation (Dhimolea, 2011).

Strikingly, combined inhibition of the inflammatory caspases, caspase-1 and caspase-11, reduced not only the number of CNV lesions that formed in VEGF-A^hyper mice (as observed when

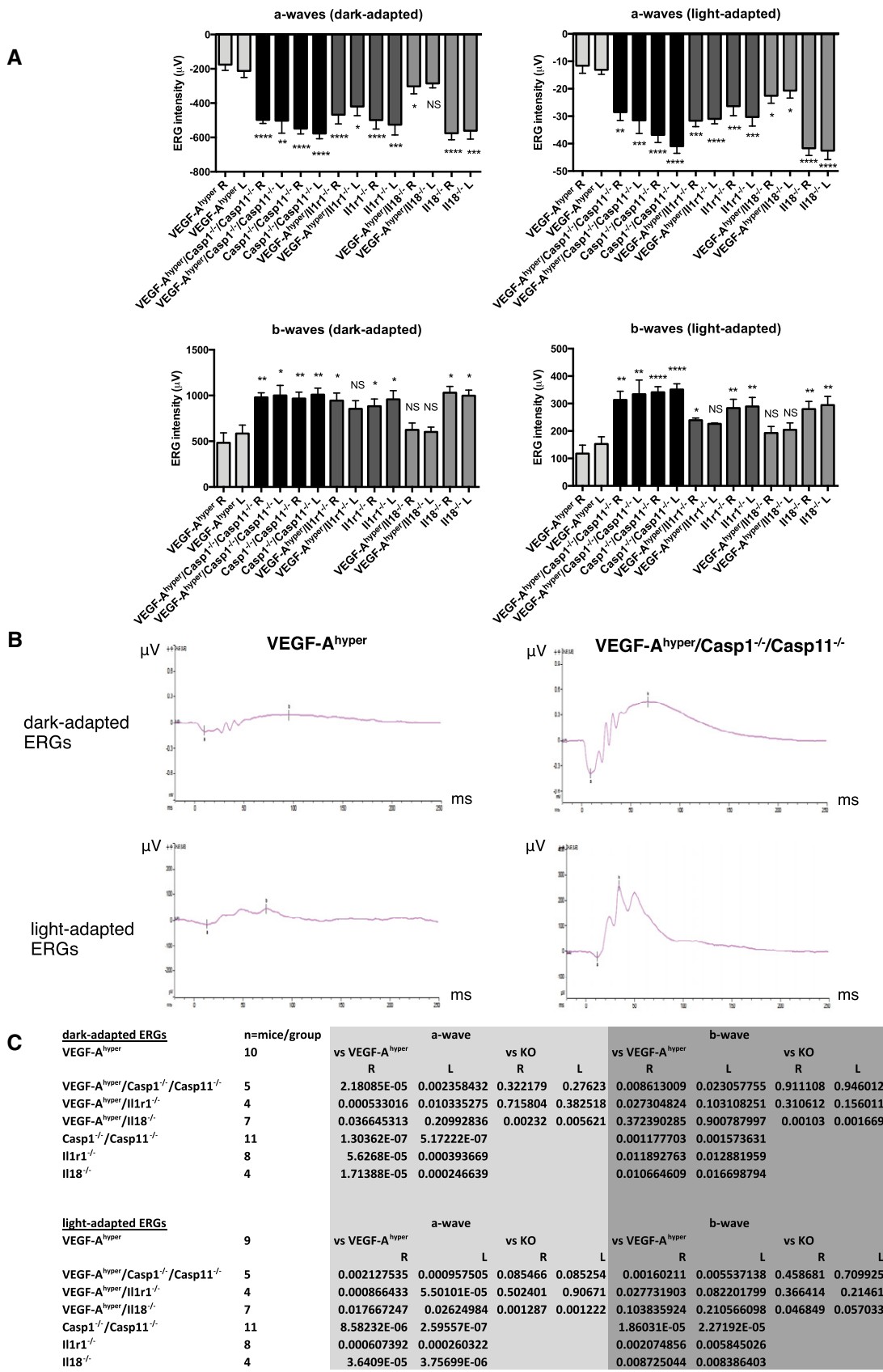

**Figure 10.**

◄

**Figure 10.   Targeting caspase-1/caspase-11 or Il1r1 partially rescues the loss of visual function in aged VEGF-A^hyper mice.**

A    Graphs show means of a- and b-waves of dark-adapted and light-adapted ERG responses (ERG intensities in μV) for both the left and the right eyes from 21-month-old VEGF-A^hyper mice, VEGF-A^hyper/Casp1^−/−/Casp11^−/− mice, Casp1^−/−/Casp11^−/− mice, VEGF-A^hyper/Il1r1^−/− mice, Il1r1^−/− mice, VEGF-A^hyper/Il18^−/− mice, and Il18^−/− mice. None of these mice had cataracts. Graphs show mean ± SEM. *P*-values are indicated in (C).

B    Representative ERG tracings are shown for VEGF-A^hyper mice and for VEGF-A^hyper/Casp1^−/−/Casp11^−/− mice. *X*-axes show milliseconds (ms). *Y*-axes show ERG intensities (in μV).

C    For each experimental mouse group, means for a- and b-waves of ERGs were calculated and *P*-values are shown for left and right eyes using a two-tailed unpaired Student's *t*-test. *P*-values of the VEGF-A^hyper mouse group are compared to those VEGF-A^hyper mice lacking either Casp1/Casp11, Il1r1, or Il18, as well as to the KO mice (Casp1^−/−/Casp11^−/− mice, Il1r1^−/− mice, or Il18^−/− mice) without the VEGF-A^hyper allele. In addition, *P*-values are shown comparing values between the KO mouse groups and those that have the VEGF-A^hyper allele. Mouse numbers for each experimental mouse group are indicated as well.

targeting Nlrp3), but also the size of CNV lesions. Moreover, targeting these caspases inhibited VEGF-A-induced degeneration of the RPE and photoreceptors and the age-dependent decline in visual function. Thus, our findings suggest that inflammatory caspases could be important novel therapeutic targets in the comprehensive treatment or prevention of both neovascular and non-exudative AMD, and indicate that targeting inflammatory caspases may inhibit AMD more potently than when targeting Nlrp3 or IL-1β signaling.

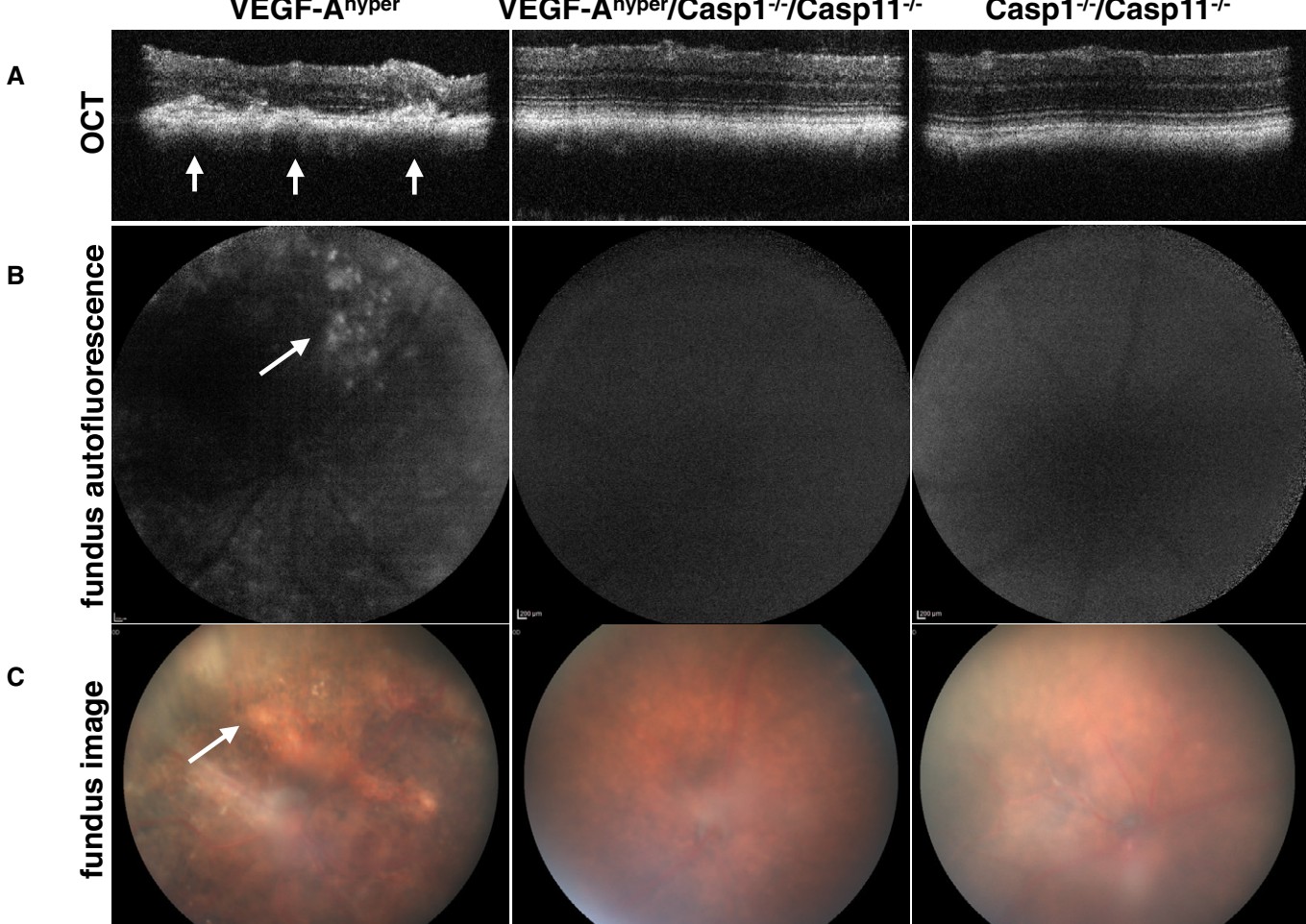

**Figure 11.   Lack of caspase-1/caspase-11 inhibits CNV lesion formation, retinal degeneration, fundus abnormalities, and fundus autofluorescence in aged VEGF-A^hyper mice.**

Representative images of 21-month-old mice are shown.

A    OCTs in aged VEGF-A^hyper mice show retinal degeneration with a thinning of the retina and CNV lesions originating from the choroid (arrows). OCTs do not show the same degree of abnormalities in age-matched VEGF-A^hyper/Casp1^−/−/Casp11^−/− mice and Casp1^−/−/Casp11^−/− mice.

B    Fundus autofluorescence imaging shows areas of autofluorescence in the fundus of aged VEGF-A^hyper mice (arrow), but not in age-matched VEGF-A^hyper/Casp1^−/−/Casp11^−/− mice and Casp1^−/−/Casp11^−/− mice.

C    Abnormal pigment loss and degenerative changes are observed in the fundi of aged VEGF-A^hyper mice (arrow), while the fundi of age-matched VEGF-A^hyper/Casp1^−/−/Casp11^−/− mice and Casp1^−/−/Casp11^−/− mice do not show these abnormalities.

Furthermore, we identified inflammasome-independent therapeutic targets for neovascular AMD by showing that targeting VEGF-A/Flk1 signaling in the RPE prevents RPE barrier breakdown and CNV, while the inhibition of TLR2 may inhibit the growth of these CNV lesions.

In summary, our data demonstrate that increased VEGF-A levels, as they can occur in aging tissues due to increased hypoxia or oxidative damage, exacerbate normal age-associated changes and may lead to the manifestation of progressive pathologies resembling those in human senile cataracts and AMD, suggesting shared pathomechanisms between these distinct age-related eye diseases. The manifestation of these common VEGF-A-induced aging eye pathologies can be strongly inhibited by targeting components of the NLRP3 inflammasome. Thus, targeting NLRP3 inflammasome components may serve as a comprehensive unifying therapeutic approach to simultaneously inhibit or prevent multiple common aging diseases of the eye, while preserving the beneficial effects of VEGF-A function.

## Materials and Methods

### Animals

For all animal studies, institutional approval by Massachusetts General Hospital was granted. Experiments were performed in compliance with ARRIVAL guidelines. The generation of VEGF-A$^{hyper}$ mice was previously reported (kindly provided by Dr. Andras Nagy) (Miquerol *et al*, 1999). Mice between ages 6 weeks to 34 months were examined, in total > 400 mutant and control littermate mice. The increase in VEGF-A levels occurs in these mice as a consequence of the heterozygous insertion of an IRES-NLS-lacZ-SV40pA sequence +202 bp 3′ to the STOP codon into the 3′-UTR of the VEGF-A gene locus. Nuclear β-galactosidase expression reflects VEGF-A expression at single-cell resolution in these mice. The described AMD-like pathologies were observed only in VEGF-A$^{hyper}$ mice, while none of the control littermate mice displayed these RPE/retinal pathologies. We quantified CNV lesions in mice at 6 weeks of age and every single VEGF-A$^{hyper}$ mouse of > 400 examined mice showed RPE barrier breakdown and CNV lesions. Thus, the fully penetrant AMD-like phenotypes in VEGF-A$^{hyper}$ mice allow for precise comparative quantifications of these eye pathologies. Furthermore, VEGF-A$^{hyper}$ mice (VEGF-A$^{lacZ/WT}$ mice) were obtained through intercrosses with their control littermates (VEGF-A$^{WT/WT}$ mice) for > 25 generations. The observed ocular abnormalities always co-segregated with the VEGF-A$^{hyper}$ allele (VEGF-A$^{lacZ/WT}$) and were not observed in any of the examined control littermate VEGF-A$^{WT/WT}$ mice, making it highly unlikely that another mutation may have contributed to the observed ocular abnormalities in VEGF-A$^{hyper}$ mice. By utilizing littermate mice (VEGF-A$^{WT/WT}$ mice from VEGF-A$^{lacZ/WT}$ intercrosses) as controls for the VEGF-A$^{hyper}$ mice, we ensured the same genetic background.

Mice in which the IRES-NLS-lacZ-SV40pA sequence was inserted immediately 3′ to the STOP codon into the 3′-UTR of the VEGF-A gene locus are hypomorphic for VEGF-A (VEGF-A$^{hypo}$ mice), but maintain β-galactosidase expression from the endogenous VEGF-A gene locus (Damert *et al*, 2002). While these mice showed β-galactosidase expression in the same cells and tissues as seen in VEGF-A$^{hyper}$ mice, the pathologies in VEGF-A$^{hyper}$ mice were not observed in these VEGF-A$^{hypo}$ mice, demonstrating that these pathologies are not caused by insertion of the lacZ sequence and expression of β-galactosidase (Fig 9I). VEGF-A$^{hyper}$ mice were crossed with Il18$^{-/-}$, Il1r1$^{-/-}$, Casp1$^{-/-}$/Casp11$^{-/-}$, Nlrp3$^{A350VneoR}$ mice, Tlr2$^{-/-}$, beclin-1$^{-/WT}$, and Atg16l1$^{hypo}$ mice (mice from JAX were on a C57BL/6 genetic background; Atg16l1$^{hypo}$ mice were kindly provided by Dr. Skip Virgin) (Glaccum *et al*, 1997; Takeda *et al*, 1998; Brydges *et al*, 2009). Homozygous Nlrp3$^{A350VneoR}$-knockin mice contain a floxed *neoR* cassette in opposite orientation in intron 2 of the Nlrp3 gene leading to the lack of functional Nlrp3 gene expression in the absence of Cre expression (here referred to as Nlrp3$^{-/-}$ mice) (Brydges *et al*, 2009). We confirmed the absence of an Nlrp3 transcript by RT–PCR with primers spanning exons 2 and 3 (exon 3 contains the NACHT domain, which is critical for the role of NLRP3 in inflammasome activation) (Appendix Fig S6). Mice deficient in caspase-1 have also an incidental caspase-11 deficiency, the murine homolog of CASP4 (referred to as Casp1$^{-/-}$/Casp11$^{-/-}$ mice), and caspase-11 has been shown to be involved in caspase-1 activation (Li *et al*, 1995; Wang *et al*, 1998; Kayagaki *et al*, 2011). None of these null mice (without the VEGF-A$^{hyper}$ allele) developed CNV lesions (> 30 mice/group were examined). To generate mice with RPE-specific overexpression of VEGF-A, we crossed Vmd2-Cre mice that express Cre recombinase driven by the RPE-specific promoter Vmd2 after P10 (from JAX) with ROSA-STOP$^{fl/fl}$-VEGF-A$^{164}$ mice (kindly provided by Dr. Andras Nagy) to generate mice that specifically overexpress VEGF-A$^{164}$ in the RPE postnatally (Vmd2-Cre$^{+/WT}$/ROSA-STOP$^{fl/fl}$-VEGF-A$^{164}$ mice) (Iacovelli *et al*, 2011). RPE-specific Cre expression was confirmed by immunolabeling for Cre. In addition, we crossed Flk1$^{fl/fl}$ mice (from JAX) with Vmd2-Cre$^{+/WT}$ mice and VEGF-A$^{hyper}$ mice to generate Vmd2-Cre$^{+/WT}$Flk1$^{fl/fl}$VEGF-A$^{hyper}$ mice. Both the *rd1* and the *rd8* mutations were excluded in all mouse strains analyzed in this study by PCR. All mutant strains were bred for > 10 generations. All mice were housed under conventional breeding conditions.

### Morphological examination of eyes

For semithin sections, eyes were fixed in 1.25% paraformaldehyde and 2.5% glutaraldehyde in 0.1 M cacodylate buffer (pH 7.4). After postfixation in 4% osmium tetroxide, and dehydration steps, tissues were embedded in TAAB epon (Marivac Ltd.), and 1-μm-thin sections were used for toluidine blue staining and light microscopy. Posterior eye sections were assessed at the level of the optic nerve, and representative retinal images were obtained from areas adjacent to the optic nerve in a paracentral location. Images from similar locations were obtained from each eye, which allowed a direct comparison of eyes between different mouse strains. Lens sections were analyzed in both plastic-embedded semithin sections, as well as in OCT-embedded frozen sections.

### Immunolabeling

Whole eyes were fixed in 4% paraformaldehyde overnight at 4°C and then washed in PBS. Thereafter, the eyes were dissected along

the ora serrata. For choroidal flat mounts, eyes were permeabilized in 0.5% Triton X-100 and blocked with 5% serum in which the secondary antibodies were raised for 1 h. For frozen sections, fixed tissues were treated with 30% sucrose and subsequently embedded in OCT and cut at 7 μm thickness (lenses were cut at 2 μm thickness). Tissue permeabilization was performed with 0.5% Triton X-100, and blocking was performed with blocking serum in which the secondary antibody was raised. Incubation with primary antibodies in blocking solution was performed overnight at 4°C and then washed in PBS. Primary antibodies used were as follows (at a 1:50 dilution): rat anti-mouse CD31 (MEC13.3, BD PharMingen), rabbit anti-mouse β-galactosidase (A11132, Life Technologies), Alexa-647-conjugated rat anti-mouse F4/80 (clone BM8, Biolegend), mouse anti-Cre (clone 2D8, Millipore), rabbit anti-NLRP3 (H66, Santa Cruz Biotechnology), rabbit anti-mouse C1qA (M-120, Santa Cruz Biotechnology), rabbit anti-mouse TLR2 (ab24192, Abcam), and rabbit anti-mouse C5b-9 (ab55811, Abcam). Fluorescein-conjugated isolectin B4 (Vector) was used for intracardiac perfusions to label perfused vasculature. Cytoskeletal staining was performed with Alexa-488-conjugated phalloidin (Life Technologies). Co-labeling experiments were combined with single-labeling experiments and experiments omitting both either the primary or the secondary antibodies to distinguish immunolabeling from autofluorescence. Secondary antibodies (at a 1:100 dilution) were conjugated Alexa-488, Alexa-555, or Alexa-647 antibodies (Life Technologies) and were incubated for 3 h at room temperature in the dark. DAPI was used for staining of nuclei (Life Technologies).

## Quantification of the size and number of CNV lesions

We used for all CNV quantifications 6-week old mice. The eyes were enucleated and fixed in 4% paraformaldehyde overnight at 4°C and then washed in PBS. The anterior segment, the lens and the retina were removed, and the RPE/choroid tissue (posterior eye) was treated with 0.5% Triton X-100 in 5% blocking buffer (with blocking serum in which the secondary antibody was raised) and subsequently used for immunolabeling with anti-CD31 antibodies to detect blood vessels, Alexa-488-conjugated phalloidin, and anti-β-galactosidase antibody to confirm lacZ expression in the RPE as a genotype control. The area of each CNV lesion (in μm$^2$) was measured using Zeiss AxioVision software and the average CNV lesion size for each eye determined. The number of CNV lesions for each eye was counted as well. Unbiased measurements were performed without knowing the genotyping information at the time of analysis. Subsequently results were assigned to the genotypes. The differences between age-matched mouse strains were determined. *P*-values were calculated with a two-tailed unpaired Student's *t*-test. Average CNV lesion number and CNV size per mouse are shown on a scatter plot (Prism 6.0b, GraphPad) (Fig 8A and B). In addition, average CNV lesion area values were log-transformed and CNV lesion number values were root-square-transformed and *P*-values calculated separately (Appendix Fig S3A and B).

## RT–PCR and semiquantitative RT–PCR

To determine VEGF-A isoform and receptor expression, RNA was isolated from the lens and RPE/choroid with TRIzol reagent (Life

Technologies). cDNA was obtained using the Transcriptor First Strand Synthesis Kit utilizing hexamer primers (Roche). cDNA was used to amplify VEGF-A isoforms and Flk1 and Flt1, and the PCR product was loaded on a 3% agarose gel. For gene expression studies, semiquantitative RT–PCR was performed using a LightCycler 480 system with the LightCycler 480 SYBR Green I master mix (Roche Applied Science, Indianapolis, IN). Primers for mouse 36B4 were used as normalization control. Concentrations were determined using a standard dilution curve. Semiquantitative RT-PCR experiments for all samples were performed in triplicate with $n = 7$ mice/experimental group. Primers for RT–PCR and semiquantitative RT–PCR are listed in Appendix Table S1.

## Measurements of serum 4-HNE and of lens reduced glutathione levels

We measured 4-HNE adducts (4-hydroxynonenal) in serum of mice, a byproduct of lipid peroxidation during oxidative stress, using a highly sensitive competitive ELISA according to the manufacturer's instructions (CellBiolabs). Reduced glutathione (the major antioxidant in the lens) in lens lysates was determined with a HT Glutathione Assay kit (Trevigen), according to the manufacturer's instructions. $N = 7$ mice/experimental group (two independent experiments).

## Analysis of cataract formation

Eyes were analyzed with a slit-lamp microscope. Cataract maturity was classified according to the extent of morphological opacification of the lens *in vivo*. Mild opacification of the lens that still allowed a proper fundus examination was classified as grade 1, while increased opacification that restricted a full fundus examination was classified as grade 2. Complete opacification of the lens with leukocoria (white pupil) represented fully matured cataracts and were classified as grade 3 cataracts.

Ratio (in %) of mice with cataracts per total mice in each age group is shown in the graphs (Fig 3A and B).

## Laser-induced CNV model and experimental treatments

Eyes of male 8-week-old C57BL/6J mice were exposed to laser photocoagulation for the induction of experimental CNV after eyes had been dilated with 1% tropicamide ($n = 15$ mice/experimental group). Laser photocoagulation was performed using a 532-nm laser (Visulas 532S; Carl Zeiss Meditec, Dublin, Ireland). Lesions were induced using a power of 120 mW, a spot size of 100 μm, and a duration of 100 ms. Eyes were assessed 7 days after laser treatment. The size of CNV lesions was measured in choroidal flat mounts. The eyes were enucleated and fixed in 4% paraformaldehyde overnight at 4°C. The anterior segment, the lens, and the retina were removed, and the RPE/choroid tissue (posterior eye) was treated with 0.5% Triton X-100 in 5% blocking buffer (with blocking serum in which the secondary antibody was raised) and subsequently used for immunolabeling. The following antibodies were used for whole mount immunolabeling of the RPE/choroid tissue: (i) anti-CD31 antibodies (rat anti-mouse CD31 (MEC13.3, BD PharMingen)) to detect blood vessels (Alexa Fluor-488 secondary antibody), (ii) anti–smooth muscle actin (SMA) antibodies [mouse monoclonal SMA-Cy3

conjugate (clone 1A4, Sigma)], and (iii) Alexa-647-conjugated rat anti-mouse F4/80 (clone BM8, Biolegend) to detect macrophages. Choroidal flat mounts were analyzed by epifluorescence microscopy using a Zeiss microscope (Carl Zeiss Microscopy, Jena, Germany) or with a Nikon confocal microscope (Nikon Instruments Inc, Melville, NY). Images were obtained with a 20× objective. CNV lesions were measured using Zeiss AxioVision software version 4.8.2. Average CNV size was determined for each mouse, and differences between mouse groups were assessed by a two-tailed unpaired Student's $t$-test. $P$-values < 0.05 were considered to be statistically significant. Exclusion criteria for CNV lesions were the formation of a retinal plug and abnormally large CNV lesions that formed over large choroidal vessels. $CD31^+$ CNV lesion area was measured.

Experimental mouse groups were treated with the caspase-1 inhibitor Ac-YVAD-cmk (25 μg/mouse caspase-1 inhibitor II in DMSO, Calbiochem) subcutaneously daily from the day of the laser treatment. This dose and administration regimen has been shown to inhibit caspase-1-mediated activities in mice (Lalor *et al*, 2011). Control mice received the same daily volume of DMSO subcutaneously. Ac-YVAD-cmk is a selective irreversible inhibitor of caspase-1 and has been shown to inhibit inflammasome activation. Additional experimental groups were treated with either the neutralizing anti-TLR2 antibody (clone T2.5, eBioscience) or an isotype-matched IgG control antibody (2 μg/mouse intravenously per tail vein injections on the day of the laser treatment and on day 2 and day 4 after laser treatment). This treatment regimen has previously been shown to inhibit TLR2 activity in mice (Wang *et al*, 2014).

### Electroretinograms (ERGs), fundus imaging, and optical coherence tomography (OCT)

Age-matched 21-month-old experimental mouse groups were used for ERG experiments. ERG recordings of both left and right eyes of each mouse were obtained. Eyes with cataracts were excluded from the analysis to distinguish retinal abnormalities from the effects of cataracts on ERG recordings.

Following overnight dark adaptation, the animals were prepared for ERG recordings under dim red light. While under anesthesia with a mixture of ketamine (120 mg/kg i.p.) and xylazine (20 mg/kg i.p.), the animal body temperature was maintained at 38°C, using an electrical heated platform, and the pupils of the mice were dilated using 1% tropicamide and 2.5% phenylephrine hydrochloride applied on the corneal surface. One drop of GenTeal eye gel (corneal lubricant) was applied to the cornea to prevent dehydration and to allow electrical contact with the recording electrode. A 25-gauge platinum needle, inserted subcutaneously in the forehead, served as the reference electrode, while a needle inserted subcutaneously near the tail served as the ground electrode. A series of flash intensities were produced by a Ganzfeld color dome controlled by the Diagnosys Espion3 (Diagnosys LLC) to test both scotopic and photopic responses.

Dark-adapted ERG recordings (to assess mainly rod function) are shown for measurements obtained after 4-ms light flashes (white light 6,500 K; 6 sweeps; 55,100-ms intersweep delay) with a light intensity of 24.1 cd.s/m$^2$.

Light-adapted ERG recordings (to assess mainly cone function) are shown for measurements obtained after 7 min of light adaptation

**The paper explained**

**Problem**

There are currently no effective therapies for "dry" AMD, and only a subset of patients with "wet" AMD respond to anti-VEGF-A therapies. Approaches to prevent the formation of senile cataracts are also very limited. Thus, novel therapeutic targets for these age-related eye diseases are urgently needed.

**Results**

We show that an increase in VEGF-A is sufficient to induce multiple common aging pathologies of the eye, resembling "wet" and "dry" AMD, as well as cataracts. Targeting components of the NLRP3 inflammasome strongly inhibits these aging eye diseases from developing. Moreover, we identify a critical role of Flk1 signaling in the RPE and of TLR2 for the manifestation of "wet" AMD-like pathologies.

**Impact**

Our findings suggest common pathomechanisms involved in age-related cataract formation and in the development of both forms of AMD, and identify novel molecular targets for a broad therapeutic approach to simultaneously inhibit these distinct common eye diseases.

(white light 6,500 K, 30 cd/m$^2$) with a light intensity of 102.4 cd.s/m$^2$ (25 sweeps, 1 Hz frequency) produced by a xenon light source. For each experimental mouse group, average values for a- and b-waves were calculated and $P$-values are shown using a two-tailed unpaired Student's $t$-test.

Fundus autofluorescence assessment, fundus imaging, and OCTs were performed as previously described (Ablonczy *et al*, 2014).

### VEGF-A ELISA

Lenses from age-matched VEGF-A$^{hyper}$ mice and control littermate mice were used for VEGF-A ELISA measurements. Freshly dissected lenses were lysed in NP-40 lysis buffer (Life Technologies) with 1 mM PMSF and protease inhibitor cocktail (Complete, Roche) using the Qiagen TissueLyser II. After centrifugation, the supernatant was assayed for VEGF-A. Supernatant VEGF-A levels were determined using a mouse VEGF-A ELISA kit according to the manufacturer's instructions (R&D Systems) and values shown are pg VEGF-A/ml lysate. Statistical significance was determined with a two-tailed unpaired Student's $t$-test.

### Statistics

An unpaired two-tailed Student's $t$-test was used for statistical analyses. $P$-values < 0.05 were considered to be statistically significant. $P$-values *< 0.05, **< 0.01, ***< 0.001, ****< 0.0001.

**Expanded View** for this article is available online.

### Acknowledgements

For some technical assistance, I would like to thank Oscar Morales and Karin Strittmatter. This work was supported by grants to A.G.M. from the NEI (R01-EY019297) and from the BrightFocus Foundation.

## Author contributions

The experiments were designed and performed by AGM. The paper was written by AGM.

## Conflict of interest

The author declares that he has no conflict of interest.

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
