## [Review Process File · EMBO Molecular Medicine]

Increased VEGF-A promotes multiple distinct aging diseases of the eye through shared pathomechanisms

Corresponding author: Alexander G. Marneros, Harvard Medical School/Massachusetts General Hospital

Review timeline:

Submission date:	06 July 2015
Editorial Decision:	07 August 2015
Revision received:	11 November 2015
Editorial Decision:	16 December 2015
Revision received:	22 December 2015
Accepted:	11 January 2016

Transaction Report:

Editor: Céline Carret and Roberto Buccione

1st Editorial Decision

07 August 2015

Thank you for the submission of your manuscript to EMBO Molecular Medicine. We have now heard back from the three Reviewers whom we asked to evaluate your manuscript.

As you will see, the issues raised are few but fundamental. Although I will not dwell into much detail, I would like to highlight the main points.

Firstly, while Reviewer 2 is more positive, Reviewers 1 and 3 are more reserved and with similar concerns, albeit with different emphasis and perspective. I identify two fundamental issues that require your action.

One is the perceived lack of pathophysiological relevance of your model and thus your findings. The second main concern is that the manuscript requires much more care and detail in overall presentation and description of observations. I would also like to mention that Reviewer 2 also raises an important issue, namely the justification for the reference to corneal neovascularization.

In aggregate the Reviewers also raise various other items of concern that require your attention.

In conclusion, while publication of the paper cannot be considered at this stage, given the potential interest of your findings and after internal discussion, we have decided to give you the opportunity

to address the above concerns.

We are thus prepared to consider a substantially revised submission, with the understanding that the Reviewers' concerns must be addressed with additional experimental data where appropriate and that acceptance of the manuscript will entail a second round of review.

The overall aim is to significantly upgrade the clinical relevance and usefulness of the dataset, which of course is of paramount importance for our title.

Please note that it is EMBO Molecular Medicine policy to allow a single round of revision only and that, therefore, acceptance or rejection of the manuscript will depend on the completeness of your responses included in the next, final version of the manuscript.

As you know, EMBO Molecular Medicine has a "scooping protection" policy, whereby similar findings that are published by others during review or revision are not a criterion for rejection. However, I do ask you to get in touch with us after three months if you have not completed your revision, to update us on the status. Please also contact us as soon as possible if similar work is published elsewhere.

EMBO Molecular Medicine now requires a complete author checklist (<http://embomolmed.embopress.org/authorguide#editorial3>) to be submitted with all revised manuscripts. Provision of the author checklist is mandatory at revision stage; The checklist is designed to enhance and standardize reporting of key information in research papers and to support reanalysis and repetition of experiments by the community. The list covers key information for figure panels and captions and focuses on statistics, the reporting of reagents, animal models and human subject-derived data, as well as guidance to optimise data accessibility.

We look forward to seeing a revised form of your manuscript as soon as possible.

***** Reviewer's comments *****

Referee #1 (Comments on Novelty/Model System):

Very interesting dry AMD-like phenotype has been shown for the VEGFA-hyper mice. Please provide a table with frequency, length, severity, etc of the deposits, RPE and PR degeneration. Clearly there are pigmentary changes evident too, which in all likelihood may indicate changes in autofluorescence. This should be commented on.

Page 7: provide ref for the following sentence '(1) oxidative stress can IL-1B can induce IL-1B activation in the lens epithelial cells in vitro'.

Referee #1 (Remarks):

The authors characterize the lens and retinal phenotype of mice expressing a '2-fold' increase in VEGFA. Appropriately they evaluate young and old mice. They present data suggesting that the common denominator for the associate age-related pathology is mediated through NLRP3 mechanisms supporting the hypothesis of targeting both VEGFA and the NLRP3 inflammasome for therapy. These concepts are not necessary novel and at least in the context of age-related macular

degeneration have been hypothesized extensively. What separates this study from others is the utilization of the VEGF overexpressing mouse model, which they, in part, published on in 2013. The mouse model has the potential of being of interest to researchers within both the lens and retina field. However prior to that greater characterization of the pathology in the mice is strongly recommended.

Additional comments are below:

Page 10: Several other groups prior to the Marneros 2013 paper have shown using chimeric mice that infiltrating macrophages contribute to development and severity of neovascular lesions. Please appropriately reference those other groups rather than just list this one paper from your group.

Are the lesions indeed choroidal in origin? The image in figure 7 panel e for example indicates the involvement of the retina and could very well be retinal angiomatous proliferation (RAP) rather than CNV. This can only be differentiated with fluorescein angiography and examination of serial sections in which the break through Bruch's can be seen. Were serial sections evaluated?

Very interesting dry AMD-like phenotype has been shown for the VEGFA-hyper mice. Please provide a table with frequency, length, severity, etc of the deposits, RPE and PR degeneration. Clearly there are pigmentary changes evident too, which in all likelihood may indicate changes in autofluorescence. This should be commented on.

Page 7: provide ref for the following sentence "(1) oxidative stress can IL-1B can induce IL-1B activation in the lens epithelial cells in vitro".

Many of the data panels do not have an associated n. Please include n for all data panels in each figure. For example Figure 3 a and b.

Figure legend 4 is missing a reference to panels b and c.

Figure 7: what is the age of onset of AMD-like pathologies. At what ages are the mice with genetic inactivation of CASP, NLRP or IL1R1 evaluated?

Minor comments:

Spell out all acronyms at first use.

Page 3, first sentence of third paragraph should be corrected (remove "With"....).

Referee #2 (Remarks):

This is a comprehensive study addressing important aging diseases of the eye. Several recommendations to improve the work:

- 1) Font sizes of some of the bar figures are very small and hard to read.
- 2) Figure 1d and 1e: provide age (young vs old) images for comparison.
- 3) Corneal neovascularization is not a common senile disease like cataract and AMD, the author needs to explain why it was investigated in this study and whether it was affected by the inhibition of NLRP3 inflammasome components.

4) It is known that anti-VEGF treatment has side effects and VEGF-A is required for normal vasculature maintenance. How about the inhibition of NLRP3 inflammasome components? Does it affect normal vasculatures?

5) Contradictory statements on Flk1 function: last sentence on page 8 vs. first two sentences on page 9.

Referee #3 (Remarks):

General comments

The author has performed an extensive study of the effects of transgenic hyper-expression of VEGFA on lens and retinal cells, finding pathological changes in both circumstances. He has then crossed these mice with a variety of gene deficient mice targeting genes which are upstream or downstream of VEGFA signalling. He has then identified some predictable and unpredictable effects based on current knowledge of VEGFA signalling.

While many of these effects eg requirement for TLR2 for pathological effects in the CNV induction are of interest, whether they reflect pathophysiological processes is not clear. Additional data demonstrating similar effects in non-transgenic models using, for example, specific inhibitors / blocking antibodies or would lend support to the physiological relevance of this work.

Specific comments

The numbers of generations used for crossing the various strains should be provided.

The fixation procedures are not clear. Are they different for the lens and the posterior segment of the eye?

The details of the immunostaining are not provided.

Examination of the experimental material was performed in a blinded manner. How precisely was this i.e. who prepared the samples and who was blinded?

Details of the primers used should be provided in the supplementary materials.

Cataract grading was done: what grading system was used (eg what is meant by a moderate cataract?) and why was grading necessary (graded lenses are not discussed in the paper)?

The histological images in Figure 1, 2, 4 and 7 are unclear; it is difficult to see the correlation between the Western blot and the histogram in Fig 2.

Similar aging changes have been reported in WT as shown for hyper VEGFA mice in Figure 7. Control data is necessary to show here.

Please find below a detailed point-by-point response to the reviewers' comments:

Reviewer's comments *****

Referee #1 (Comments on Novelty/Model System): Very interesting dry AMD-like phenotype has been shown for the VEGFA-hyper mice. Please provide a table with frequency, length, severity, etc. of the deposits, RPE and PR degeneration. Clearly there are pigmentary changes evident too, which in all likelihood may indicate changes in autofluorescence. This should be commented on.

Response: *We included images that show autofluorescence of observed sub-RPE deposits in VEGF-A^{hyper} mice (Fig 5A-B). Moreover, we performed autofluorescence fundus imaging in vivo in aged VEGF-A^{hyper} mice, which showed the presence of strongly autofluorescent deposits in the eyes of aged VEGF-A^{hyper} mice. These changes were associated with degenerative changes of the RPE with pigment abnormalities that were observed by color fundus imaging (Fig 11B-C) as well as by light and electron microscopy (Fig S2 and S3). Importantly, targeting the inflammasome (by genetic inactivation of CASP1/CASP11) strongly inhibited these age-dependent non-exudative AMD-like pathologies as well as fundus abnormalities.*

RPE and photoreceptor degeneration occurs in all VEGF-A^{hyper} mice in a progressive age-dependent manner. While young VEGF-A^{hyper} mice show already degenerative changes of the RPE and retina, VEGF-A^{hyper} mice > 12 months of age show prominent basal laminar-like sub-RPE deposits and degeneration of the RPE and photoreceptors in all mice examined (100%), while none of these findings were found in littermate control mice. Thus, RPE and photoreceptor degeneration occurs in aged VEGF-A^{hyper} mice with 100% frequency. We included this information in the manuscript. We also added

high-resolution light and electron microscopy images of these RPE and photoreceptor pathologies in the revised manuscript (Fig S2 and S3).

Page 7: provide ref for the following sentence '(1) oxidative stress can IL-1B can induce IL-1B activation in the lens epithelial cells in vitro'.

Response: *We added this reference.*

Referee #1 (Remarks): The authors characterize the lens and retinal phenotype of mice expressing a '2-fold' increase in VEGFA. Appropriately they evaluate young and old mice. They present data suggesting that the common denominator for the associate age-related pathology is mediated through NLRP3 mechanisms supporting the hypothesis of targeting both VEGFA and the NLRP3 inflammasome for therapy. These concepts are not necessary novel and at least in the context of age-related macular degeneration have been hypothesized extensively. What separates this study from others is the utilization of the VEGF overexpressing mouse model, which they, in part, published on in 2013. The mouse model has the potential of being of interest to researchers within both the lens and retina field. However prior to that greater characterization of the pathology in the mice is strongly recommended.

Response: *We have performed an extensive morphologic and functional characterization of the VEGF-A^{hyper} mice. We expanded our dataset with serial sectioning of plastic-embedded eyes of these mice up to 34 months of age (in comparison to littermate control mice). Moreover, we performed extensive electron microscopy studies of the eyes of these mice. We could show that CNV lesions strongly resemble human neovascular AMD-like pathologies, originating from the choroidal vasculature and being covered by RPE cells.*

Electron microscopy also shows sub-RPE basal laminar deposits and degenerative changes of the RPE and photoreceptors that strongly resemble findings in human non-exudative AMD. We have included a series of figures in the revised manuscript to highlight these AMD-like pathologic changes (Fig 5, S2 and S3). Moreover, we have performed ERG experiments in aged VEGF-A^{hyper} mice, which show that the observed age-related retinal pathologies lead to a loss of visual function (both rod as well as cone function) as seen in human AMD. We also included clinical imaging approaches (OCTs and fundus imaging) of RPE and retinal abnormalities in these mice to strengthen the clinicopathologic correlation with human AMD.

Additional comments are below:

Page 10: Several other groups prior to the Marneros 2013 paper have shown using chimeric mice that infiltrating macrophages contribute to development and severity of neovascular lesions. Please appropriately reference those other groups rather than just list this one paper from your group.

Response: *We have included these additional references here as well.*

Are the lesions indeed choroidal in origin? The image in figure 7 panel e for example indicates the involvement of the retina and could very well be retinal angiomatous proliferation (RAP) rather than CNV. This can only be differentiated with fluorescein angiography and examination of serial sections in which the break though Bruch's can be seen. Were serial sections evaluated?

Response: *The CNV lesions in VEGF-A^{hyper} mice are of choroidal origin. We performed serial sections that showed choroidal neovessels being separated from the overlying photoreceptor layer by RPE cells. Moreover, we utilized white VEGF-A^{hyper} mice for choroidal flat mount fluorescence labeling experiments, which allow visualization of the vessels underlying the non-pigmented RPE cells by confocal microscopy. We perfused these white VEGF-A^{hyper} mice with fluorescein-lectin, and then immunolabeled the choroidal flat mounts with anti-CD31 antibodies (staining is strongly increased in*

choroidal neovessels, allowing demarcation of neovessels from quiescent choroidal vessels), and with phalloidin (which allows us to visualize RPE cells). Confocal microscopy demonstrates that CD31⁺ neovessels in CNV lesions of the subretinal space originate from the quiescent choroidal vasculature and are covered by RPE cells that separate the neovessels from the photoreceptors (Fig 5).

This observation is also consistent with OCT imaging of CNV lesions in vivo in these mice (Fig 11A), which resemble CNV lesions in neovascular AMD.

Similarly, CNV lesions that develop in mice with RPE-specific VEGF-A overexpression (in VMD2Cre-ROSASTOP^{fl/fl}-VEGF-A¹⁶⁴ mice) originate from the choroidal vasculature, are covered by RPE cells and have no connection to the retinal vasculature (Fig 6A).

Thus, our experiments clearly demonstrate that the neovascular lesions in these genetic mouse models are of choroidal origin.

Many of the data panels do not have an associated n. Please include n for all data panels in each figure. For example Figure 3 a and b. Figure legend 4 is missing a reference to panels b and c. Figure 7: what is the age of onset of AMD-like pathologies.

Response: The requested changes were made accordingly. All absolute numbers are shown in the figure legends (for cataracts separate tables of absolute and relative numbers per group are shown in Fig S1). Figure references for Fig 4 b and c are in the manuscript.

AMD-like pathologies occur already at an early age and progress in severity with progressive age. At 6 weeks of mice all VEGF-A^{hyper} mice already developed multifocal small CNV lesions, which increased in size as they became older.

Age-of-onset of non-exudative AMD-like pathologies start at an early age, but become more prominent in mice >12-months of age. This resembles the progressive age-related disease course in human "dry" AMD. We have included here histologic images from eyes of ~2-year old mutant mice to show fully formed pathologies.

At what ages are the mice with genetic inactivation of CASP, NLRP or IL1R1 evaluated?

Response: We evaluated CNV lesions at 6 weeks of age in all mice. Evaluating CNV lesions early on allows us to quantify distinct lesions before they become too large and merge with adjacent lesions. Dry AMD-like pathologies were evaluated in 2 year-old mice. At that age AMD-like pathologies are very prominent in VEGF-A^{hyper} mice. Should inhibition of the NLRP3 inflammasome indeed have a major effect on preventing VEGF-A-induced AMD-like pathologies, then we would expect that these pathologies would be inhibited even at an advanced age when they are fully developed in VEGF-A^{hyper} mice. Indeed, we observed that even in 2-year old VEGF-A^{hyper} mice genetic inactivation of NLRP3 inflammasome components could prevent the manifestation of these AMD-like pathologies. We observed that the rescue of ocular pathologies in aged VEGF-A^{hyper} mice due to targeting the NLRP3 inflammasome also correlated with maintained visual function in these mice, as assed by ERGs in 2-year old experimental mouse groups (Fig 10). Moreover, fundus abnormalities and autofluorescent deposits were strongly inhibited by targeting the inflammasome in VEGF-A^{hyper} mice (Fig 11).

Minor comments: Spell out all acronyms at first use. Page 3, first sentence of third paragraph should be corrected (remove "With"....).

Response: We have made changes accordingly.

Referee #2 (Remarks): This is a comprehensive study addressing important aging diseases of the eye. Several recommendations to improve the work:

1)Font sizes of some of the bar figures are very small and hard to read.

Response: Figure sizes and bar figures were increased. Figures have been changed to improve the presentation of the findings.

2) Figure 1d and 1e: provide age (young vs old) images for comparison.

Response: We included images from β -gal labeling of eyes of young VEGF-A^{hyper} mice as well (Fig 1). These images show that the expression pattern of VEGF-A is not affected by age. For example, the RPE, retinal cells of the inner nuclear layer (INL), and lenticular cells express VEGF-A in young and aged mice.

3) Corneal neovascularization is not a common senile disease like cataract and AMD, the author needs to explain why it was investigated in this study and whether it was affected by the inhibition of NLRP3 inflammasome components.

Response: We agree that corneal neovascularization is not a common age-related disease and thus we have focused in this manuscript on the lens and the retina, which develop VEGF-A-induced cataracts and AMD-like pathologies in an age-dependent manner. We removed the reference to corneal neovascularization from the abstract and mentioned the occurrence of corneal neovascularization in the manuscript as an interesting additional observation in these mice. We did not observe significant VEGF-A expression in β -gal labeling experiments in corneas of VEGF-A^{hyper} mice (we have included these new data in Fig 1). Notably, we only observed corneal neovascularization in aged VEGF-A^{hyper} mice that also had developed cataracts. Thus, we speculate that corneal neovascularization is an indirect consequence of increased VEGF-A levels in the eye and associated with VEGF-A-induced cataract formation. However, not all VEGF-A^{hyper} mice that formed cataracts also had corneal neovascularization. The variability of the occurrence of corneal neovascularization in aged VEGF-A^{hyper} mice limits statistical quantifications to conclusively show in a meaningful manner whether corneal neovascularization was dependent on the NLRP3 inflammasome. We have made changes in the manuscript accordingly to explain these findings in the proper context. Should the reviewers feel that the data on corneal neovascularization do not add to the current manuscript, then we would be willing to remove these data.

4) It is known that anti-VEGF treatment has side effects and VEGF-A is required for normal vasculature maintenance. How about the inhibition of NLRP3 inflammasome components? Does it affect normal vasculatures?

Response: We did not observe significant abnormalities of the adult normal choroidal vasculature in mice lacking NLRP3, caspase-1/11, IL1R1 or IL-18. Consistent with these findings, mice lacking NLRP3, caspase-1, IL1R1 or IL18 did not show obvious vascular phenotypes or a reduced lifespan. Choroidal flat mount experiments showed that none of these control groups developed spontaneous CNV lesions, which were seen in VEGF-A^{hyper} mice (>30 age-matched control mice/group were assessed). We have added this information in the manuscript.

5) Contradictory statements on Flk1 function: last sentence on page 8 vs. first two sentences on page 9.

Response: We have made changes accordingly.

Referee #3 (Remarks): General comments The author has performed an extensive study of the effects of transgenic hyper-expression of VEGFA on lens and retinal cells, finding pathological changes in both circumstances. He has then crossed these mice with a variety of gene deficient mice targeting

genes, which are upstream or downstream of VEGFA signaling. He has then identified some predictable and unpredictable effects based on current knowledge of VEGFA signaling. While many of these effects eg requirement for TLR2 for pathological effects in the CNV induction are of interest, whether they reflect pathophysiological processes is not clear. Additional data demonstrating similar effects in non-transgenic models using, for example, specific inhibitors / blocking antibodies or would lend support to the physiological relevance of this work.

Response: *This reviewer asked for additional data to demonstrate similar effects of targeting the inflammasome or TLR2 with either chemical inhibitors or neutralizing antibodies in a non-transgenic CNV mouse model in reducing CNV growth as we have observed in VEGF-A^{hyper} mice that lack caspase-1 or TLR2 . Verification of a role of the inflammasome or TLR2 for AMD-like pathologies in an independent additional CNV mouse model would strengthen the clinical and therapeutic relevance of our findings.*

In order to address this suggestion, we have now performed laser-induced CNV experiments in mice. This in vivo CNV mouse model has been used extensively in the AMD research community as a non-transgenic model of neovascular AMD. In this model, laser-injury to the RPE/choroid induces CNV-like lesions. We tested in these laser-induced CNV experiments whether pharmacologic inhibition of the critical inflammasome component caspase-1 would reduce CNV lesion growth, as our findings in VEGF-A^{hyper} mice that lack caspase-1 would suggest. Indeed, we observed that treatment of mice with a specific chemical inhibitor of caspase-1 strongly reduced laser-induced CNV in non-transgenic wild-type mice when compared to vehicle-treated control mice. These findings provide additional evidence in a second independent experimental model that targeting the inflammasome can inhibit CNV lesion growth.

Next, we tested whether pharmacologic inhibition of TLR2 would also inhibit CNV lesion growth in this non-transgenic AMD model. We treated wild-type mice with neutralizing anti-TLR2 antibodies and control mice with isotype-matched control antibodies. We found that TLR2 neutralizing antibodies significantly inhibited CNV growth as well.

Thus, as requested by the reviewer we have now provided additional data in a non-transgenic independent mouse model of neovascular AMD, which show that therapeutic targeting of caspase-1 or TLR2 indeed inhibits CNV growth. These data support our findings in VEGF-A^{hyper} mice in which we have targeted the inflammasome or TLR2 genetically, revealing a critical role of these genes for CNV growth.

Specific comments: The numbers of generations used for crossing the various strains should be provided. The fixation procedures are not clear. Are they different for the lens and the posterior segment of the eye? The details of the immunostaining are not provided. Examination of the experimental material was performed in a blinded manner. How precisely was this i.e. who prepared the samples and who was blinded?

Response: *These detailed experimental information were added to the "Materials and Methods" section (and moved from the Supplemental Material to the Methods section of the main manuscript). Unbiased analyses were performed without knowing the genotype of each mouse at the time of the analysis.*

Details of the primers used should be provided in the supplementary materials.

Response: *Primer sequences that were used are listed now in the Supplemental Methods.*

Cataract grading was done: what grading system was used (eg what is meant by a moderate cataract?) and why was grading necessary (graded lenses are not discussed in the paper)?

Response: The detailed method used for grading cataracts has now been added in the Materials and Methods part. Cataract maturity was classified according to the extent of morphological opacification of the lens in vivo. Mild opacification of the lens that still allowed a proper fundus examination was classified as grade 1, while increased opacification that restricted a full fundus exam was classified as grade 2. Complete opacification of the lens with leukocoria (white pupil) represented fully matured cataracts and were classified as grade 3 cataracts. Age-dependent cataract formation and its severity are shown in Fig 4 and S1. Cataract grading allows for a more comprehensive evaluation of differences in cataract severity between mouse groups.

The histological images in Figure 1, 2, 4 and 7 are unclear; it is difficult to see the correlation between the Western blot and the histogram in Fig 2. Similar aging changes have been reported in WT as shown for hyper VEGFA mice in Figure 7. Control data is necessary to show here.

Response: We have improved the histological images and added additional images (also in Supplemental Figures). We have also provided additional images of age-matched control littermate mice that show that the eye abnormalities in VEGF-A^{hyper} mice are not observed in control mice with normal ocular VEGF-A levels (Fig 9A and Fig S2). We have assessed a large number of control littermate mice and we did not observe any CNV lesions in these mice (we performed choroidal flat mounts or histological sections on >200 control mice). With progressive age control mice show mild age-appropriate changes of the RPE/choroid interface, but no AMD-like pathologies were observed in these mice (Fig 9A and Fig S2).

In summary, we have performed an extensive revision of our original manuscript and included substantial additional experimental data that strengthen the clinicopathologic correlation of the findings in our mouse model with those observed in human age-related ocular pathologies. These additional experiments include laser-induced CNV experiments, detailed light and electron microscopic characterization of the ocular pathologies in these mice, as well as clinical assessment of visual function and clinical parameters of AMD (ERGs, OCTs, fundus color and autofluorescence imaging). Our observations reveal important novel pathogenic roles of the NLRP3 inflammasome in mediating VEGF-A-induced age-related ocular diseases and provide the framework for new therapeutic approaches that target NLRP3 inflammasome components in these diseases.

Thank you for the submission of your revised manuscript to EMBO Molecular Medicine. We have now received the enclosed reports from the referees that were asked to re-assess it. As you will see the reviewers are now globally supportive and I am pleased to inform you that we will be able to accept your manuscript pending the following final amendments:

1) Please address the remaining set of comments from referees 1 and 3 in writing in the main text and in a rebuttal letter

Please submit your revised manuscript within two weeks. I look forward to seeing a revised form of your manuscript as soon as possible.

***** Reviewer's comments *****

Referee #1 (Remarks):

The manuscript has been improved significantly and many of my previous concerns have been resolved.

A couple of points:

1. Figure 1, shows b-gal staining in the RPE of young mice (Figure 1B), but in the older mice 'strong VEGF staining in the RPE' is actually not seen (Figure 1G), so the statement that strong VEGF staining persists with age is not accurate. Also see, Page 11, first line. VEGFA expression is actually quite high in the inner retina rather than the RPE.
2. The increased VEGF levels in the literature is reported for aqueous samples and the text throughout the manuscript should be corrected to reflect this. Currently it is cited as VEGF levels are high in human tissue samples, without clearly stating that the 2 fold increase was actually measured in the aqueous.
3. Figure 2, shows VEGF levels from the lens, what about the RPE of these mice? Is there an increase in VEGF levels with age? Please show parallel data.

Referee #2 (Remarks):

Previous questions have been addressed. Since corneal neovascularization shows variability and is not well studied as other phenotypes, it is recommended to remove the data from this manuscript.

Referee #3 (Remarks):

I have re-checked the Figures and legend to Figure 8F. I think the legend could be more explanatory but the data are fine and impressive.. I am pleased to see these data included in the paper. I have no reservations now about the work.

December 21th, 2015

Re: EMM-2015-05613

Dear Editor,

We have now fully addressed all of the reviewers comments. Please see below a point-by-point response.

Response to Editor:

3) Figures 3A and 2B:

Some irregularities have been seen in these 2 instances, in order to assess the validity of the data, we now encourage the publication of source data, particularly for electrophoretic gels and blots, with the aim of making primary data more accessible and transparent to the reader. Would you be willing to provide a PDF file per figure that contains the original, uncropped and unprocessed scans of all or key gels used in the figure? The PDF files should be labeled with the appropriate figure/panel number (1 file/figure), and should have molecular weight markers; further annotation may be useful but is not essential. The PDF files will be published online with the article as supplementary "Source Data" files. If you have any questions regarding this just contact me.

Response:

We have enclosed the uncropped full Western blots for the ERK and ERK-P experiments that are depicted in Figure 3a. Both Western blots give a clean and clear result with little background bands. The observed bands correspond to the expected bands for ERK and ERK-P (42 and 44 kDa).

The RT-PCR results (Figure 2b) also correspond to the expected bands.

***** Reviewer's comments *****

Referee #1 (Remarks):

The manuscript has been improved significantly and many of my previous concerns have been resolved.

A couple of points:

1. Figure 1, shows b-gal staining in the RPE of young mice (Figure 1B), but in the older mice 'strong VEGF staining in the RPE' is actually not seen (Figure 1G), so the statement that strong VEGF staining persists with age is not accurate. Also see, Page 11, first line. VEGFA expression is actually quite high in the inner retina rather than the RPE.

Response:

We have made changes accordingly in the manuscript to avoid direct comparisons regarding the staining intensity of VEGF-A in the RPE/retina:

"In the retina, we found that the RPE is ~~the~~ a major cell type in the posterior eye to express VEGF-A in the adult, while ~~some~~ expression of VEGF-A was also noticed in cells of the ganglion cell layer and the inner nuclear layer (Fig 1B). This expression pattern was maintained throughout life and ~~strong~~ VEGF-A expression in the RPE was also found in > 2 years-old mice (Fig 1G)."

We have also explained that lacZ staining in these mice is nuclear:

"As VEGF-A^{hyper} mice have a NLS-lacZ cassette in the 3'UTR of the VEGF-A gene, staining for β -galactosidase can be used to detect cellular VEGF-A expression in these mice (β -galactosidase appears as nuclear staining)."

According to the reviewer's suggestion we have removed statements about the quantities of VEGF-A expression with progressive age. However, lacZ staining is nuclear and the staining observed in the RPE in aged mice is strong and fully saturated (Fig 1G, nuclei in red show fully saturated staining), and this staining appears more intense than in the retina (inner nuclear layer) in the immunofluorescence stainings of 2-year old eyes of VEGF-A^{hyper} mice.

To determine the contributions of increased VEGF-A expression specifically in the RPE we have generated mice with RPE-specific increased expression of VEGF-A and show that this increase of VEGF-A expression in the RPE is sufficient to cause CNV (Figure 6A).

2. The increased VEGF levels in the literature is reported for aqueous samples and the text throughout the manuscript should be corrected to reflect this. Currently it is cited as VEGF levels are high in human tissue samples, without clearly stating that the 2 fold increase was actually measured in the aqueous.

Response:

This was corrected accordingly:

"This increase of VEGF-A levels was maintained with age and is similar to the reported ~2-fold increase of VEGF-A in aqueous humor samples from eyes of patients with neovascular AMD..."

3. Figure 2, shows VEGF levels from the lens, what about the RPE of these mice? Is there an increase in VEGF levels with age? Please show parallel data.

Response:

We have previously reported increased VEGF-A protein levels (by ELISA) in the RPE/choroid of aged VEGF-A^{hyper} mice (Marneros, 2013, Cell Reports). We have included this reference.

Referee #2 (Remarks):

Previous questions have been addressed. Since corneal neovascularization shows variability and is not well studied as other phenotypes, it is recommended to remove the data from this manuscript.

Response:

According to the reviewer's suggestion we have removed the data on corneal neovascularization.

Referee #3 (Remarks):

I have re-checked the Figures and legend to Figure 8F. I think the legend could be more explanatory but the data are fine and impressive.. I am pleased to see these data included in the paper. I have not reservations now about the work.